# Population-wide analysis of differences in disease progression patterns in men and women

David Westergaard [1], Pope Moseley[1], Freja Karuna Hemmingsen Sørup[1,2], Pierre Baldi[3] & Søren Brunak[1]

Sex-stratified medicine is a fundamentally important, yet understudied, facet of modern medical care. A data-driven model for how to systematically analyze population-wide, longitudinal differences in hospital admissions between men and women is needed. Here, we demonstrate a systematic analysis of all diseases and disease co-occurrences in the complete Danish population using the ICD-10 and Global Burden of Disease terminologies. Incidence rates of single diagnoses are different for men and women in most cases. The age at first diagnosis is typically lower for men, compared to women. Men and women share many disease co-occurrences. However, many sex-associated incongruities not linked directly to anatomical or genomic differences are also found. Analysis of multi-step trajectories uncover differences in longitudinal patterns, for example concerning injuries and substance abuse, cancer, and osteoporosis. The results point towards the need for an increased focus on sex-stratified medicine to elucidate the origins of the socio-economic and ethological differences.

[1] Novo Nordisk Foundation Center for Protein Research, Faculty of Health and Medical Sciences, University of Copenhagen, 2200 Copenhagen, Denmark. [2] Unit of Clinical Pharmacology, Roskilde University Hospital, 4000 Roskilde, Denmark. [3] Institute for Genomics and Bioinformatics and Department of Computer Science, University of California, Irvine, CA 92697, USA. Correspondence and requests for materials should be addressed to S.B. (email: soren.brunak@cpr.ku.dk)

Sex- and gender-stratified medicine is an essential aspect of precision medicine. Sex and gender affect the manifestation and pathophysiology of many diseases[1–3]. Sex is defined as the biological component, while gender is a social construction as for example defined by the WHO[4]. Sex is a separate risk factor even when all other aspects have been taken into account[5–7]. Although sex is an important aspect of disease, many sex-specific analyses focus on one sex only and less on the comparative aspect[8]. Consequently, sex- and gender-medicine is generally understudied, and an increasing body of literature stresses the need to include both sexes in animal models, clinical trials, and healthcare planning policies[8–12]. Men and women are affected differently by disease, such as cardiovascular diseases, osteoporosis, and autoimmune diseases[2,3,13–16]. Furthermore, many prior studies also indicate a bias in diagnosis and treatment, for example that osteoporosis is underdiagnosed in men, while chronic obstructive lung disease is underdiagnosed in women[16,17]. Although earlier studies point to clear sex-specific differences in a number of disease states, they have not yet been complemented by multimorbidity studies that incorporate co-occurrence of other conditions in a systematic manner. Some co-occurring conditions display a consistent temporal progression trend. However, cross-sectional studies are time-unresolved and most cohort studies define a priori the temporal association between conditions when testing a specific hypothesis, and thus do not take into account the order in which conditions are observed in clinical care. Nonetheless, the etiology and outcome of single conditions will very often be related to their temporal context in terms of other conditions[18–20]. A temporal trend is a prerequisite for causality and should systematically be taken into consideration when studying patient-specific co-occurrences of conditions[21,22].

Incidence and temporality in diagnosis co-occurrence have been studied previously, but the focus has not been centered on sex-stratified differences[20,23]. We now present a retrospective cohort study based on the population-wide Danish National Patient Registry (NPR), where we examine sex-specific incidence, risk, and temporal aspects of diagnoses and co-occurrence of diagnosis related to disease and symptoms. Our findings indicate large discrepancies across all areas of disease.

## Results

**Diagnosis incidence and relation to age.** We analyzed hospital admissions from 6,909,676 patients (the whole Danish population during a 21-year period), of which 48.2% were women. We analyzed the incidence rate of 1369 ICD-10 level 3 diagnoses for men and women. A complementary analysis using the Global Burden of Disease (GBD) categories can be found in Supplementary Note 1. Incidence rates may be biased by age; thus we calculated the age-adjusted incidence rate (AIR) using the Eurostat 2013 standard population[24]. The Methods section contains a detailed account of the statistical model employed. We found that 344 and 473 diagnoses had a higher AIR in women and men, respectively (see Supplementary Data 1 for estimates and 95% Bayesian Credible Intervals (BCI)). Differences in incidence rates were not limited to a few particular disease areas, but distributed across the 18 ICD-10 chapters studied (Fig. 1a). Nonetheless, some ICD-10 chapters such as infectious diseases (ch1), neoplasms (ch2), circulatory system diseases (ch9), respiratory diseases (ch10), perinatal conditions (ch16), and injuries (ch19) had a higher AIR in men, on average. Conversely, endocrine and metabolic disorders (ch4), eye and adnexa diseases (ch7), skin diseases (ch12), musculoskeletal diseases (ch13), and congenital malformations (ch17) had a higher AIR in women, on average. A very similar pattern was observed when using the GBD categories

(Supplementary Figure 1A). Considering the age of first hospital diagnosis we found 986 diagnoses in which the age was different for men or women (Welch's $t$ test, FDR < 0.05) (see Supplementary Data 2 for mean values and 95% confidence intervals (CI)). We noticed that in the majority of cases, women were, on average, diagnosed at an older age than men (Fig. 1b, c). The only exceptions were neoplasms (ch2), blood and immune system diseases (ch3), and genitourinary system diseases (ch14). From the analysis using the GBD categories we also found that men were, in the majority of the cases, diagnosed at a younger age compared to women (Supplementary Figure 1B, C).

**Diagnosis co-occurrence.** Following frequency-based filtering, we analyzed 27,185 diagnosis co-occurrences, including both sex and non-sex-specific diagnoses (Fig. 2). In the analysis, we adjusted for a number of common confounding factors, including age, admission type, hospitalization month, and year by selecting a matched comparison group. Modifying disease definitions and diagnostic criteria may affect both incidence and prevalence[25]. Previous studies found that changes in diagnostic criteria increased the hospitalization rate for e.g. acute myocardial infarction (AMI), and increased the prevalence and shifted the age of diagnosis for autism[26,27]. The criteria for hospitalization year and month in our scheme negate this type of effect as well as any seasonal influence, which may change the incidence of, for instance, infectious diseases. The Methods section contains a detailed account on the statistical model (see Supplementary Data 3 for estimates and 95% BCI of relative risks and directionality).

We found 12,122 directional pairs (defined as diagnosis co-occurrences that had an elevated relative risk and preferred statistical direction), when calculating the sex-adjusted RR. Remarkably, 4155 directional pairs (2055 in men and 2100 in women) were not common to both men and women. Hence, 4155 directional pairs are driven purely by one sex. This finding could be a result of a lack of power to detect the direction in either sex, but an analysis of the number of men and women diagnosed with the 12,122 pairs showed a high correlation ($\rho = 0.861$, 95% CI 0.857–0.863, Pearson correlation) (Supplementary Figure 2). For the 4155 directional pairs only, the correlation coefficient decreased slightly ($\rho = 0.799$, 95% CI 0.788−0.81, Pearson correlation).

We performed a separate analysis of the excluded dagger −asterisk pairs and found that, overall a dagger code, the etiology, precedes an asterisk code, the manifestation (Supplementary Note 2).

When taking sex into account, we found 9547 directional pairs in men and 10,380 directional pairs in women, respectively. Of these 6885 were shared leaving 2662 and 3495 unique pairs, respectively (reduced to 2514 and 2660 when not including sex-specific diagnoses). We examined the strength in directionality of the 6885 shared pairs (Supplementary Figure 3). We found that the variances of the two distributions were not equal, and that the distribution for women had a larger variance using both the ICD-10 ($F = 0.8269$, 95% CI 0.79−0.87, $F$ test) and GBD ($F = 0.52$, 95% CI 0.42–0.66) terminologies. We noted that the distribution for women was skewed towards positive values, indicating a weaker trend in directionality compared to the sex-adjusted directionality overall. We also found that the majority of directional pairs included a nonchronic diagnosis, even when excluding the symptoms and injuries chapter (Supplementary Table 1).

To obtain an overview of the anatomical and functional differences between men and women in terms of the directional pairs identified, we investigated the distribution over the 18 ICD-

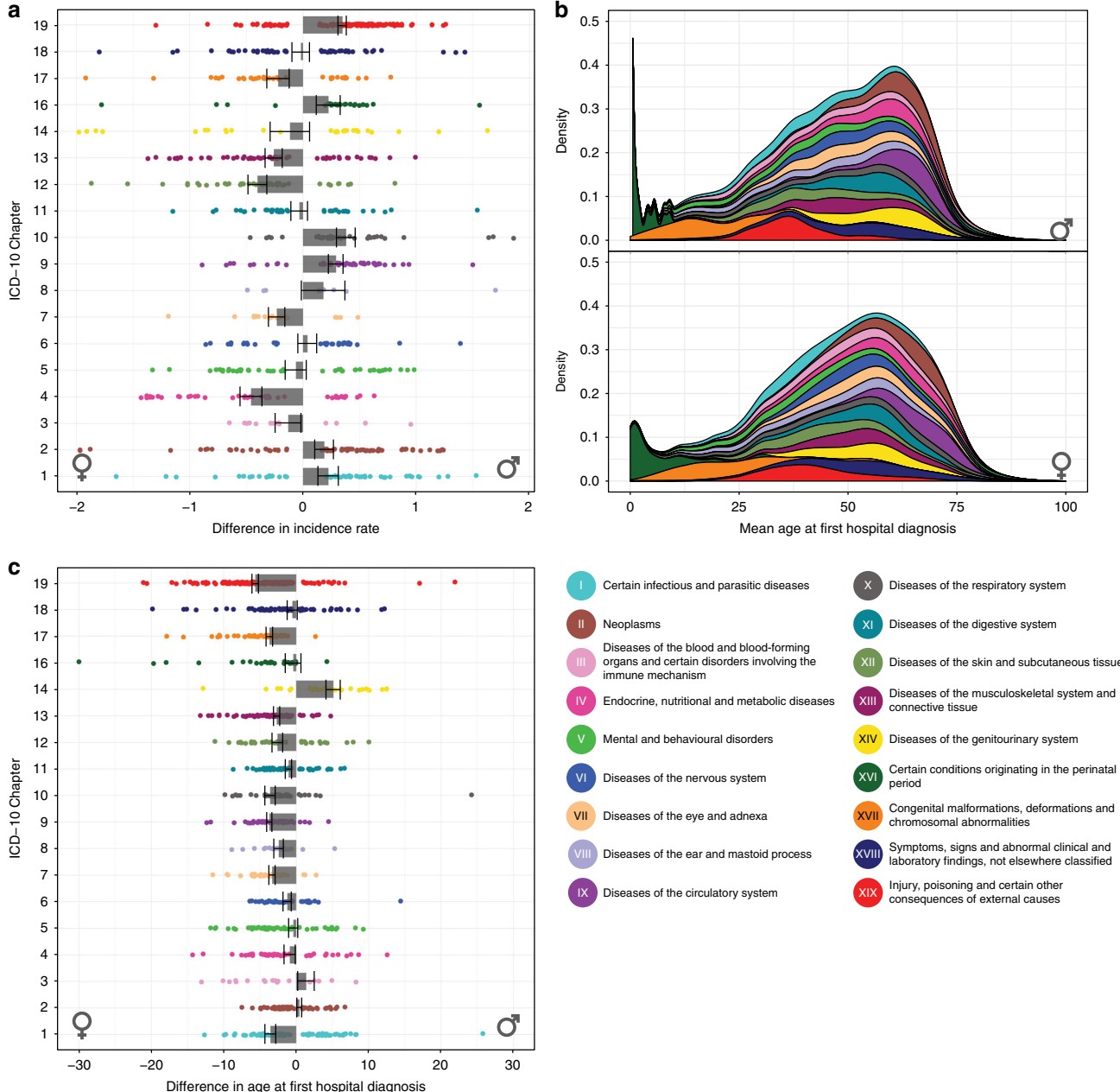

**Fig. 1** Incidence and age at first hospital diagnosis of 1369 diagnoses. **a** 344 and 437 diagnoses were found to have a higher age-adjusted incidence rate in men and women, respectively. **b** Mean age at first diagnosis for each of the 1369 diagnoses studied. **c** Mean of the difference in age at first diagnosis. We found 963 diagnoses in which the age at first diagnosis was statistically significant when comparing men and women (Welch's *t* test, FDR < 0.05). Errors bars are the standard error of the mean per ICD-10 chapter

10 chapters. We only included directional pairs identified that were unique to one sex and at the same time did not include a sex-specific diagnosis (Fig. 3). We found that diagnosis pairs from perinatal conditions (ch16) and congenital malformations (ch17) were preferentially diagnosed first in both men and women, with the exception of "neoplasms (ch2) and congenital malformations" in women. Nonetheless, there were also incongruities, such as "genitourinary system diseases (ch14) and infections (ch1)", and "neoplasms (ch2) and digestive system diseases (ch11)". Using the GBD terminology we noticed one case in which infectious diseases (ch1) were diagnosed prior to mental disorders (ch5) (Supplementary Figure 4). This was, in fact, opposite to what the analysis using the ICD-10 terminology indicated. In men, the ICD-10 terminology indicated that skin diseases (ch12) were

diagnosed prior to infectious diseases (ch1), and the opposite was found using the GBD.

Seven combinations of chapters were found to be unequally represented, hereof five overrepresented in women (FDR ≤ 0.05, Fisher's exact test) (Supplementary Table 2). Diagnoses related to "neoplasms (ch2) and digestive system diseases (ch11)", and diagnoses regarding injuries (ch19) were overrepresented in men. Diagnoses related to "infectious diseases (ch1) and musculoskeletal diseases (ch13)", "neoplasms (ch2) and circulatory system diseases (ch9)", "respiratory diseases (ch10) and signs and symptoms (ch18)", "musculoskeletal diseases (ch13) and signs and symptoms (ch18)", and "musculoskeletal diseases (ch13) and circulatory system diseases (ch9)" were overrepresented in women.

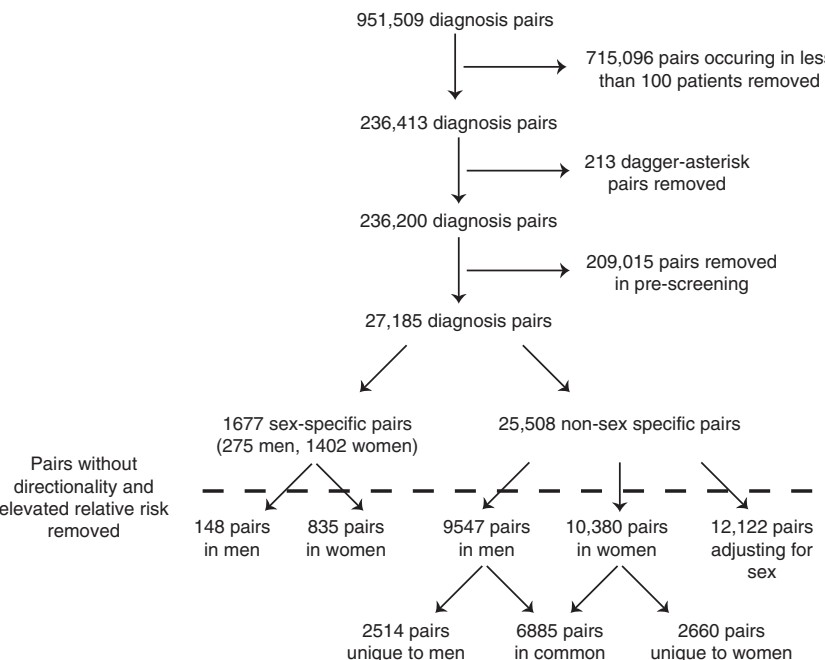

**Fig. 2** Diagnosis co-occurrences found in population-wide data from 6,909,676 patients. 951,509 ICD-10 level 3 diagnosis pairs were found to occur in the population; of these, a large number were filtered out due to low frequency (N < 100), dagger−asterisk combinations, or due to not passing the crude estimate of the relative risk. The standard method for calculating a confidence interval was applied in the prescreening section. Post-filtering 27,185 diagnosis pairs remained comprising 1360 unique diagnoses. Of these, 275 pairs involved a male-specific diagnosis and 1402 a female-specific diagnosis

Risk factors, in this case an earlier diagnosis, may predispose men and women to some diseases unequally. We found 939 pairs where the relative risk of a future diagnosis was higher in one sex, compared to the other. We only examined pairs in which more than five men or women had been diagnosed with the two diagnoses in the preferred statistical direction. In 517 cases, women were at a higher risk, while men in 422 cases were at a higher risk (Supplementary Figure 5A). We identified several inconsistencies, such as "mental disorders (ch5) and neoplasms (ch2)", in which the overall trend for the chapters were in the opposite order. When we examined the distribution of ICD-10 chapters to which the event, i.e. the diagnosis following the exposure, belonged, we found nine chapters that were unevenly represented: endocrine and metabolic disorders (ch4), mental disorders (ch5), eye and Adnexa diseases (ch7), digestive system diseases (ch11), skin diseases (ch12), and musculoskeletal diseases (ch13) in men, while ear and mastoid diseases (ch8), respiratory diseases (ch10), and genitourinary system diseases (ch14) were overrepresented in women (FDR ≤ 0.05, Fisher's exact test) (Supplementary Table 3, Supplementary Figure 6A). We compared 302 of these findings to earlier reports by searching for mentions of both ICD-10 terms in PubMed and Google Scholar. Full text articles were inspected for evidence or mentions of sex-specific risk. In total, we found solid evidence for 42 co-occurrences in which there had been reported a difference between men and women (Supplementary Dataset 9). Of these, 33 articles agreed with our findings, five provided only weak evidence by mentions of sex as a risk factor and no quantitative estimate or reference. Four articles reported opposite conclusions. These four articles were based on cohort sizes ranging from 83 to 74,020 individuals. We noticed that the directional pairs with the largest difference in relative risk from the GBD analysis was centered on substance abuse and retroviral diseases, and disorders of psychological development. Additionally, we found that men with chronic obstructive pulmonary disease (COPD) were at a

higher risk of lower respiratory infections and other respiratory disorders (Supplementary Data 7).

Inspecting the median time difference between the first occurrences, we found that there were 1181 directional pairs in which the timespan was different in men and women (FDR ≤ 0.05, Mann−Whitney U test) (Supplementary Figure 5B). In 851 of these, the time-spans between the two diagnoses were higher in women, compared to men. Here three chapters were over-represented: respiratory diseases (ch10) in women, and circulatory system diseases (ch9) and injuries (ch19) in men (Supplementary Table 4, Supplementary Figure 6B).

At the extreme, the temporal relationship between exposure and event (e.g. diagnosis A and diagnosis B) may be reversed for men and women. This reversal could point to physiological or etiological differences or may also reflect diagnostic biases within the healthcare system. For example, our overall analysis indicated that ischemic heart disease (IHD, I25) precedes paroxysmal tachycardia (PT, I47). While this pattern holds for men, it is reversed in women; IHD precedes PT in men, and PT precedes IHD in women. Thus, men mediate the observed order of occurrence at a population-wide level (Fig. 4a). We identified 15 pairs using the ICD-10 terminology and one pair using the GBD terminology in which this reversal occurs, according to our criteria (Table 1). In ten cases, there were no preferred statistical direction at the population level, while the sex-specific preferred statistical direction was reversed. In the remaining five cases, the overall preferred statistical direction corresponded to the trend in men. In some cases, the pairs involved a chronic disease and a complication of this disease. Men were diagnosed with abscess of anal and rectal regions (K61) followed by Crohn's disease (K50) in 56 out of 100 cases, where women were diagnosed in the same order in 44 out of 100 cases (Fig. 4b). Eight of the reversed pairs describe conditions related to the bladder and kidney. From the GBD analysis we identified one relationship in which the order of diagnosis was reversed, namely "pancreatitis" and "gallbladder

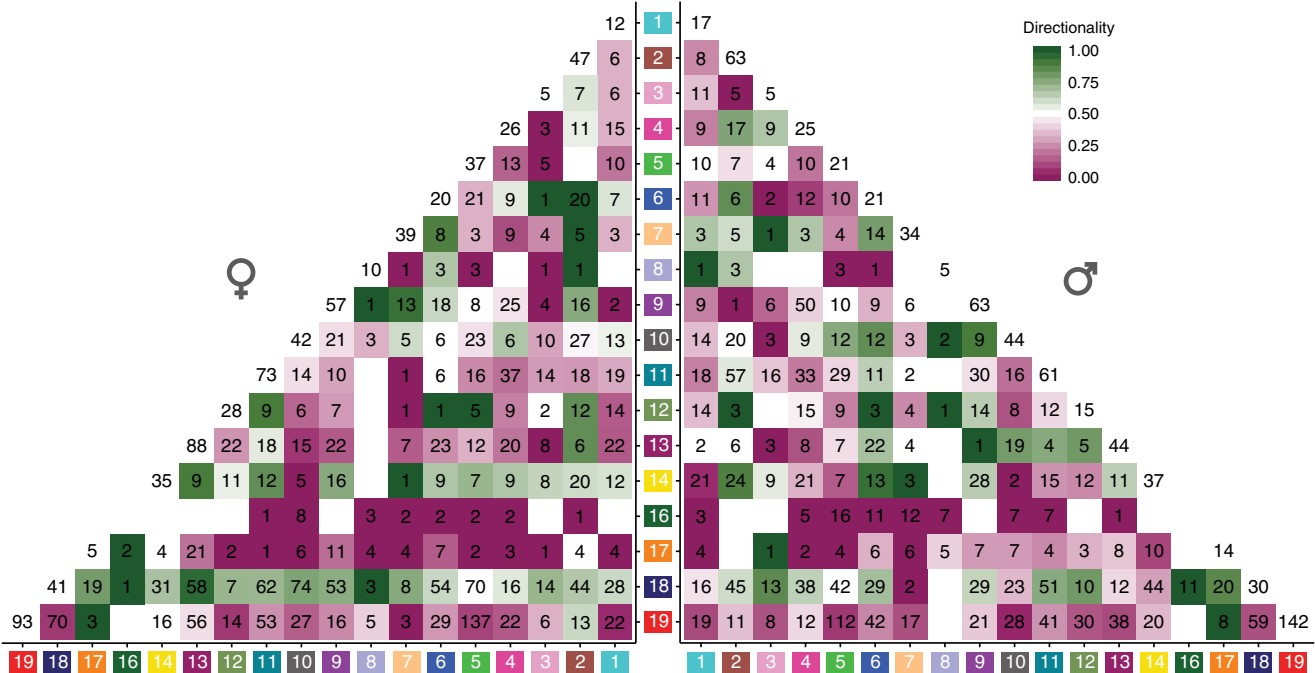

**Fig. 3** Temporal diagnosis co-occurrence across ICD-10 chapters. The distribution of 3186 and 3721 temporal diagnosis co-occurrences across ICD-10 chapters in men and women, respectively (non-sex-specific diagnoses). The color scale indicates the percentage of the pairs that has the temporal directionality from the horizontal chapter to the vertical chapter. Numbers in the boxes indicate the breakdown of the overall co-occurrence figures

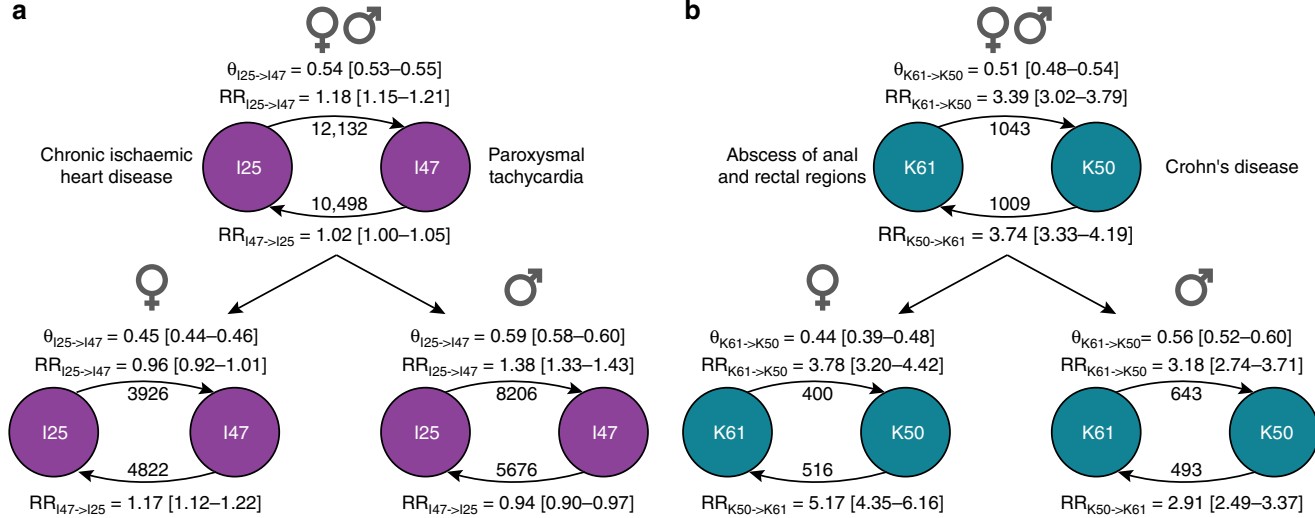

**Fig. 4** Opposite temporal relationships in men and women. **a** At the population level, paroxysmal tachycardia (I47) is observed to be a complication of ischemic heart disease (I25). The sex-stratified analysis showed that this pattern only existed in men, and that the reversed pattern was significant in women. **b** At the population level, there was no preferred direction of diagnoses between Crohn's disease (K61) and abscesses of anal and rectal regions (K50). However, the sex-stratified analysis found that the directionality was reversed between men and women

and biliary diseases". In men, "pancreatitis" was diagnosed prior to "gallbladder and biliary diseases", whereas the reverse was found in women. The directionality observed at the population level corresponded to that in men.

**Diagnosis trajectories**. Piecing together individual directional pairs may point towards overseen patterns and sex-related differences in a more extended temporal context. One framework for investing this is diagnosis trajectories[19]. We investigated the

ten directional pairs (defined as a diagnosis co-occurrence with increased relative risk and preferred statistical direction) with the largest difference in relative risk between men and women. We found 230 linear diagnosis trajectories containing at least four diagnoses (followed by at least 100 patients) (Table 2). Clustering the individual trajectories together into one trajectory network that display one directional pair only once, we noticed that there were large disparities in diagnoses related to cancers, injuries, and drug and alcohol abuse (Supplementary Fig. 7). Several diagnoses related to fractures and injuries lead to or from alcohol abuse-

**Table 1 Reversed directional comorbidities (including 95% BCI)**

| A | B | Name | Name | N, Men | N, Women | RR, men | RR, women | Direction, men | Direction, women |
|---|---|------|------|--------|----------|---------|-----------|----------------|------------------|
| R33 | N30 | Retention of urine | Cystitis | 24,519 | 8214 | 2.18 [2.12-2.24] | 1.82 [1.74-1.92] | 0.61 [0.60-0.62] | 0.42 [0.41-0.44] |
| I48 | I47 | Atrial fibrillation and flutter | Paroxysmal tachycardia | 22,648 | 18,865 | 3.29 [3.18-3.40] | 2.53 [2.46-2.61] | 0.56 [0.55-0.57] | 0.48 [0.47-0.49] |
| R33 | R31 | Retention of urine | Unspecified hematuria | 21,612 | 2302 | 1.66 [1.61-1.71] | 1.17 [1.07-1.26] | 0.52 [0.51-0.53] | 0.45 [0.42-0.48] |
| K30 | R10 | Dyspepsia | Abdominal and pelvic pain | 15,145 | 30,544 | 1.32 [1.28-1.36] | 1.41 [1.38-1.45] | 0.53 [0.52-0.54] | 0.47 [0.47-0.48] |
| I25 | I47 | Chronic ischemic heart disease | Paroxysmal tachycardia | 13,882 | 8748 | 1.38 [1.33-1.43] | 1.17 [1.12-1.22] | 0.59 [0.58-0.60] | 0.45 [0.44-0.46] |
| N31 | N30 | Neuromuscular dysfunction of bladder, not elsewhere classified | Cystitis | 3551 | 3158 | 2.35 [2.18-2.53] | 2.53 [2.32-2.76] | 0.55 [0.53-0.57] | 0.46 [0.44-0.49] |
| H26 | H27 | Other cataract | Other disorders of lens | 1000 | 1376 | 3.74 [3.23-4.30] | 2.03 [1.82-2.25] | 0.60 [0.55-0.64] | 0.40 [0.37-0.44] |
| R33 | N39 | Retention of urine | Other disorders of urinary system | 16,860 | 7012 | 2.24 [2.16-2.32] | 1.70 [1.61-1.79] | 0.66 [0.65-0.67] | 0.44 [0.42-0.45] |
| J34 | J32 | Other disorders of nose and nasal sinuses | Chronic sinusitis | 2330 | 1538 | 3.36 [3.00-3.77] | 6.47 [5.65-7.45] | 0.54 [0.51-0.57] | 0.41 [0.38-0.45] |
| R31 | N10 | Unspecified hematuria | Acute tubulo-interstitial nephritis | 2486 | 2345 | 1.61 [1.48-1.76] | 1.33 [1.21-1.45] | 0.57 [0.54-0.60] | 0.44 [0.41-0.47] |
| K61 | K50 | Abscess of anal and rectal regions | Crohn's disease [regional enteritis] | 1136 | 916 | 3.18 [2.74-3.71] | 5.17 [4.35-6.16] | 0.56 [0.52-0.60] | 0.44 [0.39-0.48] |
| N32 | N39 | Other disorders of bladder | Other disorders of urinary system | 1500 | 3212 | 1.80 [1.61-2.00] | 3.72 [3.41-4.07] | 0.57 [0.54-0.61] | 0.41 [0.39-0.44] |
| R39 | N39 | Other symptoms and signs involving the urinary system | Other disorders of urinary system | 12,044 | 6109 | 1.58 [1.51-1.65] | 2.22 [2.10-2.36] | 0.54 [0.53-0.55] | 0.39 [0.37-0.40] |
| R33 | R32 | Retention of urine | Unspecified urinary incontinence | 3189 | 3199 | 1.72 [1.58-1.86] | 2.31 [2.13-2.50] | 0.55 [0.53-0.58] | 0.40 [0.37-0.42] |
| S00 | S02 | Superficial injury of head | Fracture of skull and facial bones | 33,893 | 15,328 | 1.37 [1.34-1.40] | 1.53 [1.48-1.58] | 0.53 [0.52-0.53] | 0.47 [0.46-0.49] |
| B5.9 | B5.8 | Pancreatitis | Gallbladder and biliary diseases | 2441 | 832 | 2.46 [2.31-2.61] | 3.59 [3.39-3.8] | 0.55 [0.53-0.56] | 0.63 [0.61-0.64] |

BCI Bayesian Credible Interval, RR relative risk

related codes, with a higher relative risk in women. Moreover, women have a higher relative risk of hepatic failure following esophageal varices (Fig. 5). Second, cancers with well-known sex differences, such as thyroid cancer, bladder cancer and breast cancer are apparent (Fig. 6). The trajectory analysis was based on the most extreme directional pairs only; other diagnosis trajectories will result from including pairs with more moderate effect. For instance, there is a well-known connection between obstructive lung disease and osteoporosis[28]. Using the disease trajectory framework, we investigated the disease progression pattern in men and women (Fig. 7). We observed that obstructive lung diseases prior to an osteoporosis diagnosis tend to only occur in men, with the exception of asthma and acute bronchitis. Moreover, osteoporosis without fracture followed by osteoporosis

with fracture occurred only in women. Upon further inspection, we observed that this was due to the pair not having a preferred directionality in men, but that the relative risk was still elevated.

Putting these findings together, we focused on three areas of disease, in which we highlight specific differences (respiratory disorders, environmental disorders, and sarcoidosis) (Supplementary Note 3).

## Discussion

This analysis identified sex-mediated temporal differences across nearly all major disease areas. Using data from a complete population with free and equal access to high-quality healthcare, we report sex-specific AIR, age of first hospital diagnosis, diagnosis co-occurrence, difference in risk, and timespan between

**Table 2 Directional pairs used to construct linear trajectories**

| A | B | Name | Name | N, men | N, women | RR, women | RR, men | RR, difference |
|---|---|------|------|--------|----------|-----------|---------|----------------|
| D30 | D09 | Benign neoplasm of urinary organs | Carcinoma in situ of other and unspecified sites | 252 | 1170 | 53.264 | 29.55 | −23.658 |
| K40 | K41 | Inguinal hernia | Femoral hernia | 1679 | 1481 | 27.889 | 7.806 | −20.076 |
| E04 | C73 | Other nontoxic goiter | Malignant neoplasm of thyroid gland | 1613 | 446 | 24.114 | 43.091 | 18.858 |
| K70 | I85 | Alcoholic liver disease | Esophageal varices | 1168 | 2679 | 61.142 | 43.625 | −17.391 |
| D30 | C67 | Benign neoplasm of urinary organs | Malignant neoplasm of bladder | 1126 | 3886 | 26.634 | 11.128 | −15.5 |
| N63 | D24 | Unspecified lump in breast | Benign neoplasm of breast | 79 | 7068 | 4.792 | 17.193 | 12.395 |
| I85 | K72 | Esophageal varices | Hepatic failure, not elsewhere classified | 539 | 1256 | 35.411 | 23.41 | −11.883 |
| D35 | E89 | Benign neoplasm of other and unspecified endocrine glands | Postprocedural endocrine and metabolic disorders, not elsewhere classified | 349 | 332 | 10.276 | 22.048 | 11.763 |
| E66 | E68 | Obesity | Sequelae of hyperalimentation | 4684 | 813 | 7.46 | 18.541 | 11.071 |
| D41 | C67 | Neoplasm of uncertain or unknown behavior of urinary organs | Malignant neoplasm of bladder | 274 | 713 | 22.004 | 12.372 | −9.584 |

diagnosis using two complementary terminologies[29]. We found that more than half of the ICD-10 diagnoses examined had a different AIR in men and women, and this percentage was even higher using the GBD categories. The age at first hospital diagnosis was, on average, higher in women, across nearly all areas of disease. We showed that population-level estimates of the relative risk, and even directionality, often were driven by a single sex. Specifically, the jointly observed longitudinal patterns were most strongly driven by men, and the strength of directionality was weaker in women, irrespective of the terminology used. There were many non-sex-specific diagnosis co-occurrences only found in men or women; these discrepancies were tied to differences in the relative risk as well as the timespan between two diagnoses. Using the diagnosis trajectory approach, we illustrate how the sex-specific statistics can be used in the search for differences in longitudinal patterns. In three case stories within respiratory disorders, environmental disorders, and sarcoidosis we highlighted how the methods applied in this article provide insight into gender-specific trends in diseases and disease progression. Taken together this is, to our knowledge, the most comprehensive analysis of sex incongruities in a single population presented so far.

The study used a national patient registry, containing information from all private and public hospital admissions in Denmark, including all age groups. The population of Denmark is reasonably homogenous (~11.1% immigrants and descendants in 2015, of which 6.2% are from non-European countries)[30]. Thus, we expect that our observations are not confounded by race. Due to the nature of registry data, there are many latent factors for which we could not account. We have attempted to eliminate confounding from age, admission type, changing diagnostic criteria, and seasonal influence. One of the largest limitations of the study is the quality of data recording, and we cannot rule out that some of the incongruities could be explained by systematic errors. Nonetheless, the registry data are used for hospital reimbursement, undergoing yearly compensation adjustments, and thus the accuracy of most diagnoses is high[31]. We chose to only investigate the first occurrence of a diagnosis. It is extremely difficult to determine when a diagnosis is a recurrence, or just repeated due to the patient changing wards (or similar). Often, for nonacute conditions, there are waiting lists at the hospitals. Waiting times fluctuate over the 20-year period, due to political decisions on budgets, prioritization of disease areas like cancer, and new technologies. Hence, we did not include recurrences because it could potentially introduce bias and spurious findings. Other limitations regarding true disease state may be due to systematic gaps in medical evaluation, resulting in under- or overdiagnosis. This under- or overdiagnosis may result from a variety of causes, and the interaction between under-, overdiagnosis, and sex is of general interest, but not something we explored.

We used two different terminologies to examine sex differences. The ICD-10 terminology reflects the current clinical practice, and how hospital admissions have been coded since 1994 in Denmark. In a tradeoff between power and specificity, we worked with ICD-10 at the third level. The GBD categories represent clinical entities, and sometimes follow different definitions. For instance, in our analysis we would not have identified the relationship between two of the underlying components of COPD, emphysema and bronchitis, and osteoporosis had we only used the GBD terminology. Nonetheless, the GBD categories also pointed to important findings that could not be identified using only the ICD-10 terminology, such as alcoholic cardiomyopathy. Some sex-specific co-occurrences may also be treatment provoked. There is an increasing focus on sex-mediated side effects, which may be due to physical, hormonal, or even genetic differences[32,33]. This area was not an aspect we could explore further either due to lack of full access to medication data. In the co-occurrence analysis, we did not apply prior knowledge to assign the direction of association, i.e. whether diagnosis A was a risk factor of B or vice versa, but used advanced statistical models to infer the most likely order. Many conditions develop asymptomatically or with diffuse symptoms. Symptoms will often, but not always, be identified prior to the underlying cause. As a consequence, some conditions are not necessarily discovered in the order they arise. However, we do not discern whether this relates to different etiology, differences in presentation of symptoms, genetics (e.g. the well-known fact that the Y-chromosome increases the risk for CVD in men[34–36]), differences in drug usage (e.g. higher rates of cytochrome P450 CYP3A substrate metabolism in women[32]), or biases in the healthcare system (e.g. frequency of contact). The main goal was to present an overall view of sex differences, irrespective of mechanistic molecular causes, links to differences in environmental exposures, or biases in the healthcare system.

Menopause may also confound the results. This is not a condition that is recorded in the registry, but could be explored by selecting a fixed age. An earlier study found the average age to be 49 years, but the standard deviation was approximately 15[37]. Thus, selecting a fixed age could lead to a big bias, due to the large spread. This is better explored in another resource where it is explicitly recorded, e.g. the UK Biobank[38]. In cases with rare incidence or co-occurrence of diagnoses it can be difficult to

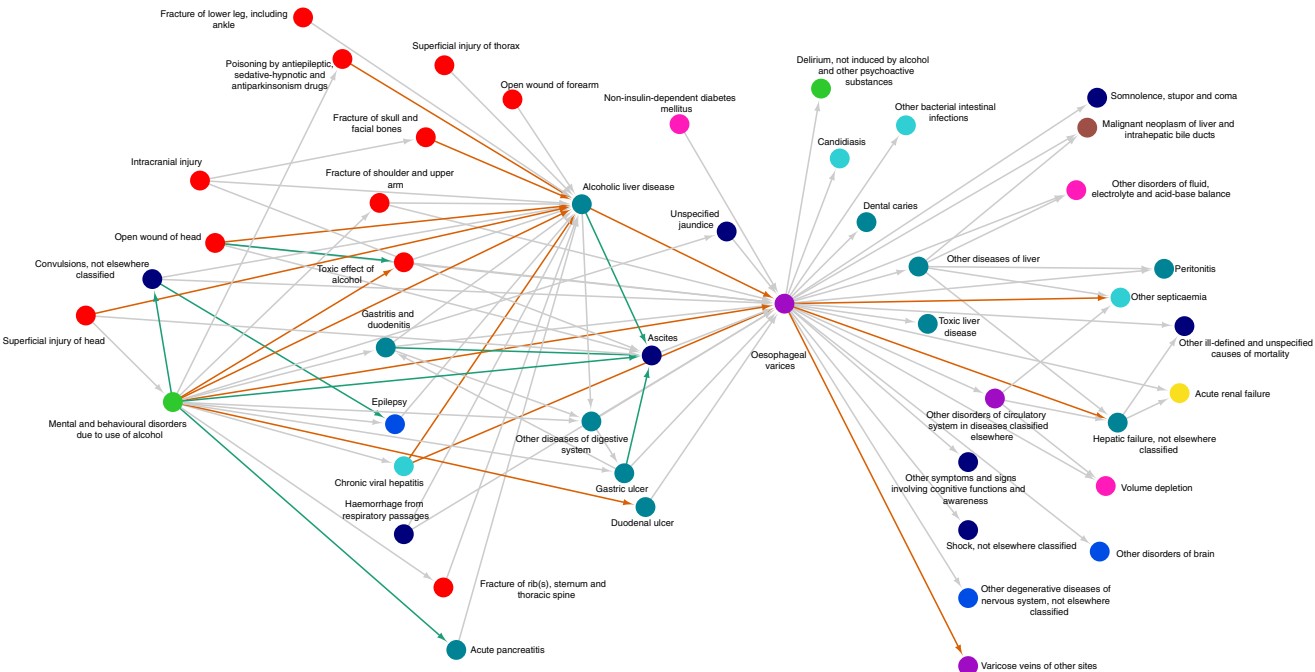

**Fig. 5** Diagnosis trajectories involving injury or drug and alcohol abuse. A trajectory network combining 176 linear diagnosis trajectories related to alcohol and substance abuse (ten directional pairs with extreme differences in relative risk). Edges represent the connection between the diagnoses with directional co-occurrence. The orange and green edges between nodes indicate co-occurrences where the RR was elevated in women and men, respectively. The RR of injuries followed by alcoholic liver disease is increased in women. Furthermore, women have a higher RR of complications following esophageal varices, such as hepatic failure. RR relative risk

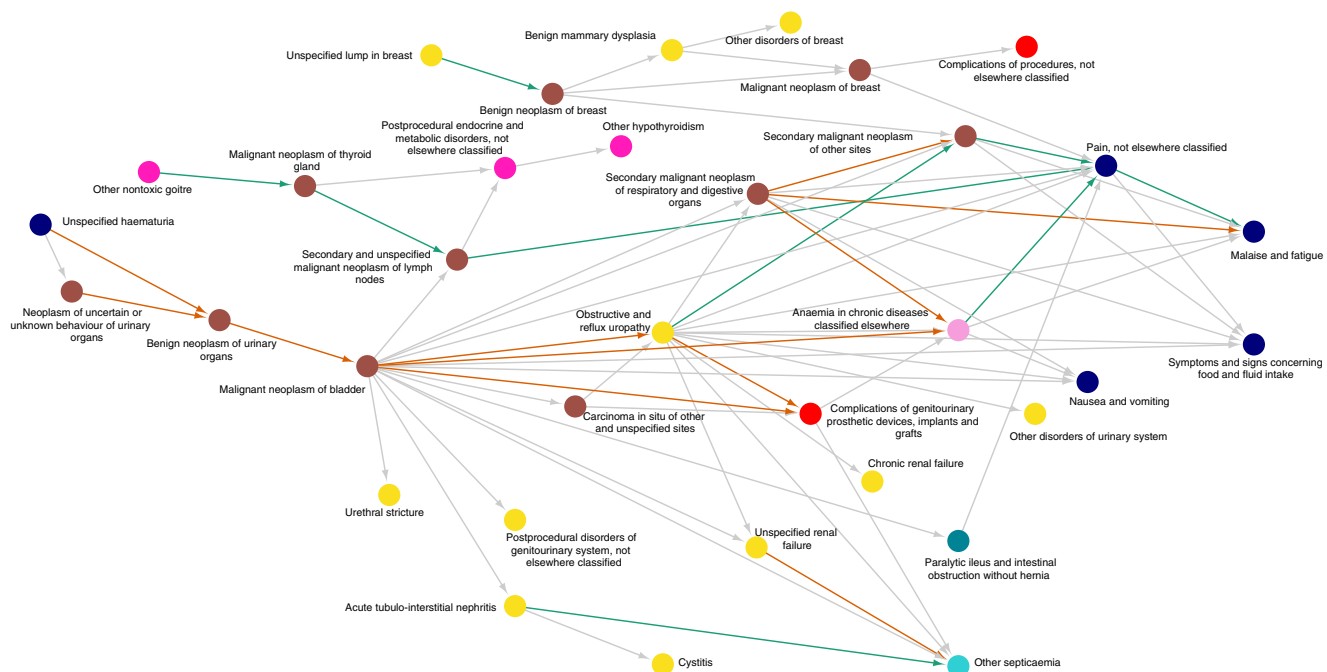

**Fig. 6** Diagnosis trajectories related to cancer. A trajectory network combining 62 linear diagnosis trajectories related to cancer (the ten directional pairs with extreme differences in relative risk). The trajectories illustrate disease routes that are related to cancers in the thyroid gland and urinary tract. The progression pattern includes secondary neoplasms, renal complications, and sepsis. Color scale as in Fig. 5

obtain a proper estimate of the standard error (SE), leading to inflated intervals for incidence rates or relative risks. We have attempted to mitigate this by using a Bayesian Hierarchical Model (BHM). The BHM improves the estimate of the SE by pooling information across groups; an approach also used in the GBD and even clinical trials[39,40]. An argument against Bayesian statistics is often that the choice of priors may introduce biases in the estimates. Conversely, here we have chosen informative priors that center the estimates at no effect, and pool the standard deviation. Thus, instead of introducing an unwanted bias, we have actually

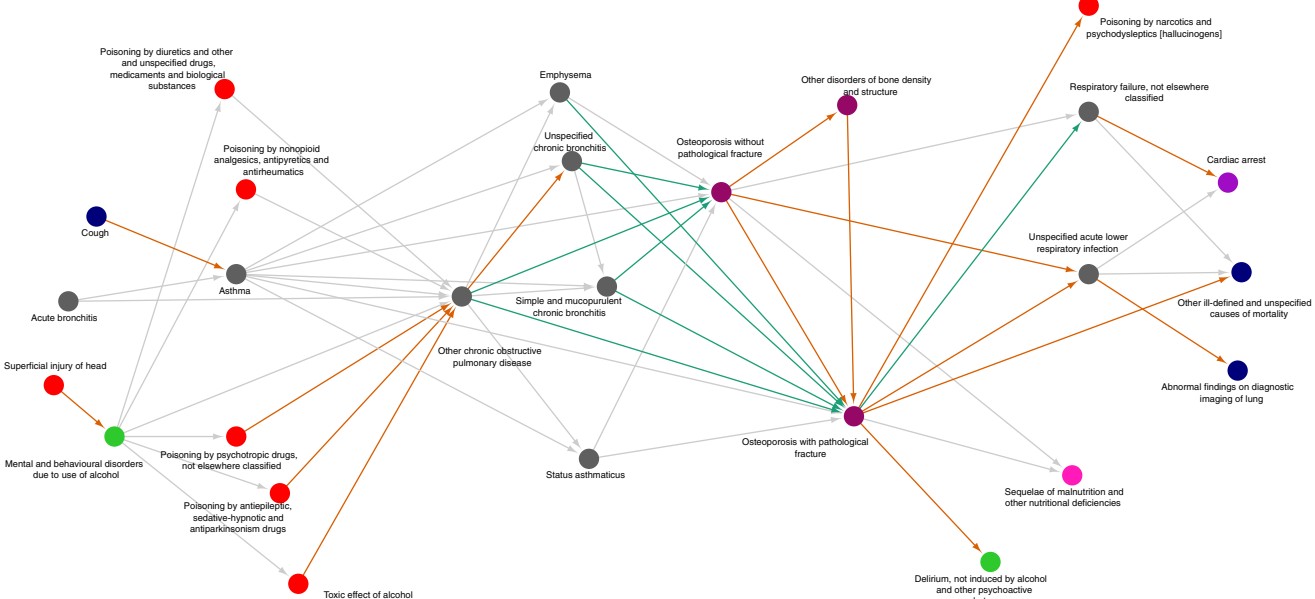

**Fig. 7** Diagnosis trajectories related to obstructive lung disease and osteoporosis. A trajectory network combining 112 linear diagnosis trajectories including osteoporosis (M80, M81) and obstructive lung diseases (J40−J46). The orange edges indicate co-occurrences only present in women, and the green edges indicate co-occurrences only present in men. The trajectories illustrate how obstructive lung diseases are found as a risk factor for osteoporosis in men, but not women. Moreover, osteoporosis without fracture followed by osteoporosis with fracture was only found in women

made a more conservative estimate compared to traditional models, which often assume an uninformative prior[41]. Lastly, we have disclosed all investigations we have performed in the Supplementary Information and provide a rich set of aggregate data that can be used in future studies.

We validated a number of the co-occurrences in the existing literature. The majority of the articles investigating the same conditions agreed with our findings. Nevertheless, this task is challenging as no other study is as broad as the one we present. Many studies do not investigate if there is a difference in sex-specific risks[8]. This omission included both meta-analysis, cohort-, and case-control studies. Sex is an important factor in epidemiological studies. In studies of single or few diagnoses with different cohorts, as well as the GBD, it is well documented that there are sex-mediated differences in the incidence rates[3,39]. Our results derived directly from hospital admissions for single disease incidence align well with previously reported differences, such as cancer, musculoskeletal disorders, and autoimmune diseases[15,42,43]. We found that the age of first hospital diagnosis was, on average, nearly always higher in women. To our knowledge, this has not been systematically studied before, and only reported for few specific areas, such as cardiovascular disorders[44]. A growing body of literature suggests that the reason for the delayed onset of cardiovascular disorders in women is due to the protective effect from estrogen[44,45]. While the age of first hospital diagnosis should not be confused with the age of onset, there is growing evidence that the protective role of estrogen is more widespread than previously thought. For instance, estrogen has been suggested to be a neuroprotective factor, which is in agreement with our findings concerning a later age of first hospital diagnosis in women for nervous system disorders[46]. Sex can also be a strong confounding factor when estimating diagnosis co-occurrence. To date no study has yet performed a systematic investigation of sex-specific diagnosis co-occurrences. We show how population-wide estimates of co-occurrence can be driven by a single sex, even when using a matched comparison group to negate other confounding factors. Furthermore, we demonstrated that the jointly observed longitudinal patterns are most strongly

driven by men, and that the strength of directionality is weaker in women. The tendency to report sex-specific estimates is becoming increasingly standard practice, in particular due to the recognized fact that sex and gender considerations are vital in precision medicine[47,48].

We found many directional pairs that were unique to one sex. In this regard, our data demonstrate that disease co-occurrences related to cancers, digestive disorders, and injuries were over-represented in men. This result indicates that men are more burdened by cancers, and complications, in the digestive system. In a temporal context, we also noticed that men are diagnosed with digestive system diseases (ch11) prior to neoplasms (ch2). This points towards a disparity in life-style-related diseases. Taken together, these findings suggest a bias in clinical practice, in which men with digestive system disorders are monitored more closely for neoplasms, whereas women are not. Men suffer more co-occurring injures. In women, both respiratory and musculoskeletal disorders were overrepresented in combination with symptoms and signs (chapter 18). One explanation for this could be that the prevalence of musculoskeletal disorders is higher in women, which leads to more unspecific symptoms, such as pain. A previous study found that women report musculoskeletal-related pain more often, and that this could be caused by a musculoskeletal sex difference[49]. In contrast to earlier large studies pooling data from multiple cohorts, we have been able to compare the timespan between temporal co-occurrences. We identified cases in which the temporal pairs had a different timespan in men and women. We note that in 72% of cases the diagnosis-free interval is longer for women than men. This finding aligns well with our earlier finding that the age of first hospital diagnoses is nearly always greater in women, and clearly shows how this widespread effect even translates into a temporal context.

The diagnosis trajectory analysis showed an increased risk in women between several injuries, substance abuse, and complications of substance abuse that we speculate could be indicators of a gender bias reflecting domestic violence and consequences from drug abuse, in light of an earlier finding that found

substance abuse to be a risk factor for nonfatal injuries in women[50]. The trajectory analysis also demonstrated a temporal relationship between nontoxic goiter, thyroid cancer, and secondary cancer in which men were at a higher risk. Women have a higher incidence of thyroid cancer, and male sex is described as a risk factor for malignant thyroid nodules. Earlier studies have found that the aggressive subtypes of thyroid cancer have a similar incidence in men and women, but that men often present at a more advanced stage[51,52]. This observation is an important finding from both epidemiological studies and this population-wide analysis and demonstrates the necessity of investigating multistep temporal associations. Furthermore, the analysis of obstructive lung disease and osteoporosis trajectories indicated patterns of severe under diagnosis. First, obstructive lung diseases were observed as risk factors for osteoporosis only in men. These results are in contrast to an earlier cross-sectional study, which found that sex did not modulate the association between airflow obstruction and osteoporosis[53]. However, the temporal relative risk may be more informative than the odds-ratio for the non-temporal co-occurrence. Moreover, obstructive lung diseases are underdiagnosed in women, a factor that can affect the estimates in a cohort study. Secondly, there was no directionality observed between osteoporosis without fracture and osteoporosis with fracture in men, but an elevated relative risk in both directions. Contrary, women were observed to have this pattern. This suggests that osteoporosis in men is not diagnosed prior to fracture, and therefore not managed. This could be part of the reason why mortality is higher in men with osteoporotic fracture, compared to women[54]. Lowered bone mineral density is a known adverse effect from corticosteroid therapy, a drug often used in the treatment of asthma and COPD. Possibly the lack of a connection for women could be due to the large difference in age of diagnosis for asthma, and therefore treatment is started later. In the case of COPD, corticosteroid therapy is only suggested for shorter symptomatic periods. However, COPD is a substantially under-diagnosed disease and two studies have estimated that 50–80% of COPD patients are undiagnosed[55,56]. Moreover, the COPD diagnosis is only confirmed by spirometry in 50% of the diagnosed patients[57,58]. Hence, patients receiving a diagnosis of COPD may be more symptomatically severe and would be expected to receive a more systemic steroid exposure. Recent data also suggest that moderate to severe emphysema itself is a risk factor for osteoporosis[59]. Another equally valid explanation could also be that the COPD phenotype carries risk of osteoporosis due to COPD associated frailty, smoking effects on bone metabolism, and limitations in physical activity. In addition, there is an interesting and emerging set of studies showing vitamin D receptor polymorphisms in patients with COPD and osteoporosis[60]. The relative impact of these factors would be greatest in men, given the baseline higher (>4 times) level of osteoporosis in women compared to men by age 50.

Our case story regarding respiratory disorders also highlighted that some complications, such as bronchiectasis and emphysema, were different in men and women, a finding that may be relevant to the clinical assessment and management. Lastly, we found 16 cases where the directionality between two diagnoses was opposite. Some of these point to conditions in which men are not diagnosed prior to serious complications, such as the case with Crohn's disease and abscesses of anal and rectal regions. Other examples included IHD and PT, and pancreatitis and gallbladder and biliary diseases. One study found that pancreatitis in men was typically alcohol induced, while in women it was due to biliary problems, which could explain the reversed order of diagnosis[61]. We speculate that the observed difference between IHD and PT, in which IHD is a recognized risk factor, could possibly be due to an under diagnosis of IHD in women. This is further complicated by the fact that men and women develop different subtypes of IHD[44].

Taken together, our findings strongly suggest many disparities in a population with a uniform, one-payer-based healthcare, again underscoring the need for better sex-stratified medicine.

Generally, many of our findings align well with larger meta studies, such as the GBD. Our study adds the dimension of the temporal aspect between disorders. In doing so, provide guidance in the design of future studies while also pointing to potential gaps in disease surveillance, diagnosis, and management. Nonetheless, a clear extension would be to perform this study in other cohorts, such as the UK Biobank although it is not comparable in size[38]. Including resources such as the UK Biobank or the emerging FinnGen and AllofUs data sets would potentially make it possible to identify genetic variants that could explain part of the discrepancy in disease progression.

## Methods

**Study design and participants**. This was a population-based registry study based on the Danish National Patient Registry (DNPR). The DNPR covered all public and private hospital admissions in Denmark during 1994–2015, 6,909,676 patients (ICD-10 period only). The healthcare system in Denmark is universal, meaning everyone living in Denmark has free access to care. Patients can be tracked through the healthcare system using the Central Person Registry (CPR) number, which is a unique identifier assigned to every Danish citizen at birth or immigration (initiated in 1968). Visits to the general practitioner (GP) and private specialist clinics were not included in the data set. Admissions included inpatient (patients admitted to the hospital overnight), outpatient (patients not admitted to the hospital overnight), and emergency department contacts. Prior to 2002 there were both full-day inpatients and half-day inpatients. After 2002, the two groups were merged into one. Hence, we merged full-day inpatient and half-day inpatient from before 2002 into one group, inpatient. Inpatient records cover the time from admission of a patient to a hospital ward, until discharge to another ward or from the hospital. If a patient was discharged to another ward, the records were combined into one record. Likewise, if the patient was re-admitted to the hospital the next day the records were combined. The data also included open outpatient contacts. If a patient has regular follow-ups at the hospital the contact may remain indefinitely as an outpatient. Since 2000, the DNPR has been used for reimbursement and the reimbursement rates are adjusted on an annual basis[62]. All referral diagnoses were excluded Referral diagnoses are used when patients are referred to another ward or department for further investigation based on a suspicion of a disorder. The ICD-10 is structured hierarchically with four levels. We studied diagnosis codes at the third ICD-10 level. We excluded ICD-10 diagnoses coming from chapters 20, 21, 22, as well as all codes specific to the Danish version of ICD-10. Codes specific for Denmark mainly describe length and weight at birth. We used the Chronic Condition Indicator to differentiate between acute and chronic ICD-10 codes (https://www.hcup-us.ahrq.gov/toolssoftware/chronic_icd10/chronic_icd10.jsp, last visited 13 July 2018). We performed a complementary analysis using the GBD categories, retrieved from http://ghdx.healthdata.org/record/global-burden-disease-study-2016-gbd-2016-causes-death-and-nonfatal-causes-mapped-icd-codes (last accessed 12 June 2018). The corresponding analysis is described in detail in Supplementary Note 1.

**Bayesian inference and model fitting**. Posterior distributions are summarized as a BCI. The BCI is the interval that spans the most credible values of the distribution, sometimes also referred to as the Highest Density Interval[63]. We defined the range of the BCI in this work to be the interval that spans 95% of the posterior distribution. Unless otherwise specified, the reported effect size is the median of the posterior distribution. We also defined a Region Of Equivalent Practice (ROPE) for the quantities of interest. A ROPE is a small region of values considered to be practically equivalent to a null value[63]. This is to ensure that the effect size of interest has a magnitude of clinical relevance, and is not just marginally different from the null value. All Bayesian models were made using the No-U-Turn sampler, a Hamiltonian Monte Carlo (HMC) variant, implemented in Stan v. 2.17.0, an open-source probabilistic programming language[64,65]. Unless otherwise specified, we ran four HMC chains, with default settings, for a total of 4000 samples, 2000 of them for warm-up to adapt HMC-specific hyper-parameters. The number of samples is significantly lower than what is usually drawn using e.g. Gibbs sampling. This is due to the nature of the NUTS-HMC algorithm, which converges faster[65]. We assessed convergence by inspecting the R-hat statistic, tree depth, and number of divergences[66,67]. The R-hat statistic describes the variation between chains. If all chains have arrived at the exact same posterior distribution for the given parameter, the R-hat will be 1. The tree depth plot is a method for assessing pathology in the HMC algorithm. If the tree depth goes to the maximal at every iteration past warm-up this indicates a random-walk behavior, which can lead to biases in the parameter estimates. A divergence happens when the model has numerical

problems (e.g. division by zero, under flowing, or over flowing), and may indicate a problematic posterior or model that does not fit the data well. In this work, we conclude that a model has converged if and only if, (1) all R-hat values are below <1.1, (2) the tree depth is not at the maximal in any of the chains past warm-up, (3) there are zero divergences.

**Diagnosis incidence rates**. We examined all diagnoses at the ICD-10 level 3 that occurred in at least 100 patients during the 21-year period. The cutoff was set to avoid diagnoses used only very rarely or never. A number of diagnoses can only occur in one sex. For instance, hyperplasia of prostate can only occur in men. To identify sex-specific diagnoses we manually curated each diagnosis examined in this work, and classified whether the diagnoses were sex specific or not (Supplementary Data 8). A trained clinician oversaw and verified the curation. To estimate the incidence, we fitted a hierarchical Bayesian Poisson model of the form shown in Eq. (1),

$$y_i \sim \text{Poisson}(\exp(\eta_i)) \tag{1}$$

in which $\eta_i$ is a linear combination of the strata for every diagnosis as shown in Eq. (2),

$$\eta_i = \beta_{i,0} + \beta_{i,\text{Age}} * x_{i,\text{age}} + \beta_{i,\text{sex}} * x_{i,sex} + \log(\text{offset}_i) \tag{2}$$

in which the age is one of the 21 5-year interval groups defined in the European Standard Population 2013 (Eurostat)[24], the sex is a binary indicator, and the offset is the population at risk. To complete the model, we specify a set of priors on the coefficients shown in Eqs. (3–5)

$$\beta_{i,0} \sim N(0, \sigma_0), \tag{3}$$

$$\beta_{i,\text{sex}} \sim N(0, \sigma_{\text{sex}}), \tag{4}$$

$$\beta_{i,\text{age}} \sim N\left(0, \sigma_{\text{age}}\right) \tag{5}$$

in which $\beta_{\text{age}}$ represents a coefficient for each of the 21 age groups, with an individual prior, $\sigma_{\text{age},}$ on each coefficient. We defined the prior on the scales of the coefficients as shown in Eqs. (6–8),

$$\sigma_0 \sim N_+(0, 3), \tag{6}$$

$$\sigma_{\text{sex}} \sim N_+(0, 0.5), \tag{7}$$

$$\sigma_{\text{age}} \sim N_+(0, 1) \tag{8}$$

in which $N_+$ is the truncated normal distribution. For the coefficients, we choose weakly informative priors, with the exception of $\beta_0$. This was due to the large number of people at risk, i.e. the offset. Therefore, we choose a less informative prior for the scale. From the fitted model, we simulated the number of cases for each diagnosis, displayed in Eq. (9),

$$\hat{y}_i = \text{Poisson}(\hat{\eta}_i) \tag{9}$$

in which the mean, $\hat{\eta}$, was equal to the estimated coefficients, Eq. (10),

$$\hat{\eta}_i = \hat{\beta}_{i,0} + \hat{\beta}_{i,\text{Age}} * x_{i,\text{age}} + \hat{\beta}_{i,\text{sex}} * x_{i,sex} + \log(\text{offset}_i). \tag{10}$$

From the fitted coefficients, we calculated the age-adjusted IR (AIR) using the European Standard Population 2013[24], as shown in Eq. (11)

$$\text{AAIR} = \frac{\sum_i p_i * N_i}{\sum_i N_i} \tag{11}$$

in which $p_i$ is the age-specific rate, and $N_i$ is the population of age group $i$, according to the European Standard Population 2013. Rates were calculated for all, men, and women, using the European Standard Population 2013, age-adjusted rates are per 100,000. If the relative difference is greater than 0.1, we conclude that there is a difference in incidence rate. The relative difference is defined in Eq. (12),

$$d = \frac{\text{AAIR}_{\text{men}} - \text{AAIR}_{\text{women}}}{(\text{AAIR}_{\text{men}} + \text{AAIR}_{\text{women}})/2}, \tag{12}$$

where a positive number will indicate a higher AIR in men, and a negative number a lower AIR in women.

**Age of first hospital diagnosis**. We calculated the average age of diagnosis for a given ICD-10 code by calculating the mean across all cases in the NPR separately for men and women. We identified differences using the Welch $t$ test. $P$ values were adjusted using the stringent Benjamini-Hochberg (BH) procedure. We report the difference in means. We estimated the chapter-wise difference by calculating the weighted mean.

**Diagnosis co-occurrence**. We examined all pairs of diagnosis that occurred in more than 100 individuals. The cutoff was set to ensure that the combination of two diagnoses is sufficiently prevalent to be of interest. The time resolution of the NPR is one day, and any diagnoses given on the same day were not counted. Only the first occurrence of a diagnosis was considered. The time of diagnosis was taken as the time the patient was discharged. If the patient had not yet been discharged, the date of the last diagnosis was used instead. ICD-10 has a dual coding system, the dagger−asterisk system. The asterisk represents the symptom or manifestation of disease and the dagger indicates the etiology of the disease. We identified these pairs and excluded them from subsequent analysis.

To negate the most common confounding factors, we sampled a matched comparison group. For any given combination of diagnoses, A and B, we fixate A as the Exposure (Ex) and B as the Event (Ev) to estimate the time-resolved relative risk, RR (A → B), and directionality, Pr (A → B). For every exposed patient, we sampled five nonexposed cases matched to (i) be in the same age group, (ii) have a hospital discharge from the same type of encounter (inpatient, outpatient, emergency department), (iii) be discharged at the same month of the same year, ±3 months. An earlier study found that the hospital encounter is a confounding factor in as much as 15% of the identified diagnosis co-occurrences from a study in the NPR[20]. Moreover, modifying disease definitions and diagnostic criteria may affect both incidence and prevalence[25]. Previous studies using NPR found that changes in diagnostic criteria increased hospitalization rate for AMI, and increased the prevalence and shifted the age of diagnosis for autism[26,27]. We negate this effect by matching the encounter year. Lastly, by matching the encounter month we diminish seasonal variation that may influence the incidence of, for instance, infectious diseases.

The relative risk is not symmetrical, i.e. RR(A → B) ≠ RR(B → A), and thus we repeat the process of selecting matched controls by fixing B as the exposure, and A as the event. This effectively doubles the number of combinations of diagnoses examined.

HMC models are computationally expensive to fit. Consequently, prior to running the full hierarchical Bayesian model using Stan we applied a prefilter by calculating the 95% CI of the relative risk using the formula provided by Morris and Gardner[68]. The relative risk is given in Eq. (13),

$$\text{RR} = \frac{N_{A \to B}/(N_{A \to B} + N_A)}{N_B/(N_B + N_0)} \tag{13}$$

and the standard error of the log-transformed RR is shown in Eq. (14),

$$\text{SE}(\log \text{RR}) = \sqrt{\left(\frac{1}{N_{A \to B}} + \frac{1}{N_{A \to B} + N_A} + \frac{1}{N_B} - \frac{1}{N_B + N_0}\right)} \tag{14}$$

hence the CI of the RR is given in Eq. (15),

$$\exp\left(\log \text{RR} \pm \left(N_{1-\alpha/2} * \text{SE}(\log R)\right)\right). \tag{15}$$

We calculated CI separately for men, women, and the two sexes combined. Only pairs of diagnoses in which either the lower bound of the CI RR(A → B) or RR (B → A) excluded 1.01 were included in the subsequent analysis, that is we only studied diagnosis co-occurrences in which the exposure increased the risk of the subsequent event by more than 1%. We note that we do not perform any correction for multiple testing. Consequently, the number of false positives will be high. Additionally, in cases with a low number of patients, the estimate of the standard error will be inaccurate. However, in the following part we describe a BHM to refine the estimate of the relative risk.

We refine the estimate of the temporal relative risk and directionality between pairs of diagnoses by employing a hierarchical Bayesian model. For each exposure, $i$, and the event observed together with this exposure, $j$, we describe the relationship using a Poisson model following Eq. (16),

$$y_{ij} \sim \text{Poisson}(\exp(\eta_{ij})), \tag{16}$$

where $\eta_{ij}$ is a linear combination shown in Eq. (17),

$$\eta_{ij} = \beta_{ij,0} + \beta_{ij,\text{Ex}} * x_{ij,\text{Ex}} + \beta_{ij,\text{Ev}} * x_{ij,\text{Ev}} + \beta_{ij,\text{ExEv}} * x_{ij,\text{ExEv}} + \log(\text{offset}_{ij}) \tag{17}$$

in which $x_{\text{Ex}}$ and $x_{\text{Ev}}$ are indicator variables for the exposure and event, respectively. $x_{\text{ExEv}}$ is the interaction between the exposure and event. The offset is

the number of people within the group. We further estimated sex-specific relative risks by introducing a sex term and interaction terms between Ex, Ev, and Sex shown in Eq. (18).

$$
\begin{aligned}
\eta_{ij} = {}& \beta_{ij,0} + \beta_{ij,\text{Sex}} * x_{ij,\text{Sex}} + \beta_{ij,\text{Ex}} * x_{ij,\text{Ex}} + \beta_{ij,\text{Ev}} * x_{ij,\text{Ev}} + \beta_{ij,\text{ExSex}} * x_{ij,\text{ExSex}} \\
& + \beta_{ij,\text{EvSex}} * x_{ij,\text{EvSex}} + \beta_{ij,\text{ExEv}} * x_{ij,\text{ExEv}} + \beta_{ij,\text{EvExSex}} * x_{ij,\text{EvExSex}} + \log(\text{offset}_{ij})
\end{aligned}
$$
(18)

To complete the model, we specify a set of priors for the regression coefficients, Eqs. (19)–(26)

$$\beta_{ij,0} \sim N(0, \sigma_0),$$
(19)

$$\beta_{ij,\text{Ex}} \sim N(0, \sigma_{\text{Ex}}),$$
(20)

$$\beta_{ij,\text{Ev}} \sim N(0, \sigma_{\text{Ev}}),$$
(21)

$$\beta_{ij,\text{EvEx}} \sim N(0, \sigma_{\text{EvEx}}),$$
(22)

$$\beta_{ij,\text{sex}} \sim N(0, \sigma_{\text{sex}}),$$
(23)

$$\beta_{ij,\text{ExSex}} \sim N(0, \sigma_{\text{ExSex}}),$$
(24)

$$\beta_{ij,\text{EvSex}} \sim N(0, \sigma_{\text{EvSex}}),$$
(25)

$$\beta_{i,j,\text{ExEvSex}} \sim N(0, \sigma_{\text{ExEvSex}})$$
(26)

and weakly informative priors on the scale of each coefficient, Eqs. (27)–(32)

$$\sigma_0 \sim N_+(0, 2),$$
(27)

$$\sigma_{\text{B}} \sim N_+(0, 2),$$
(28)

$$\sigma_{\text{ExEv}} \sim N_+(0, 2),$$
(29)

$$\sigma_{\text{EvSex}} \sim N_+(0, 2),$$
(30)

$$\sigma_{\text{ExSex}} \sim N_+(0, 2),$$
(31)

$$\sigma_{\text{ExEvSex}} \sim N_+(0, 2)$$
(32)

in which $N_+$ is the truncated normal distribution. We have removed confounding from age, admission type, admission year, and admission month by selecting five matched patients. Thus, we have not included these terms in the model, as the goal is to study the effects from sex. The prior values chosen for the interaction terms favors effects close to zero. Hence, by prior design we expect that only few of the pairs investigated will occur together more than expected by chance. In addition, the hierarchical structure imposes shrinkage on the coefficients and helps inform coefficient estimates across pairs where counts may be low[63].

Using the posterior distribution, we estimate the directionality and relative risks. We simulated the number of patients who had been diagnosed in the order A → B and B → A, and calculated the probability of observing two diagnoses in a specific direction using the formula specified in Eq. (33),

$$\Pr(A \to B) = \frac{N_{A \to B}}{N_{A \to B} + N_{B \to A}}.$$
(33)

The probability of $\Pr(A \to B)$ is thus as specified in Eq. (34),

$$\Pr(B \to A) = 1 - \Pr(A \to B).$$
(34)

We define a ROPE in the interval (0.49, 0.51). If the BCI excludes these values, we conclude that the pair of diagnoses has a preferred statistical direction. This probability can also be interpreted quantitatively. For instance, $\Pr(A \to B) = 0.8$ would correspond to A being diagnosed before B in four out of five cases.

Likewise, from the posterior distribution, we calculate an adjusted relative risk using the Cochran−Mantel−Haenszel method shown in Eq. (35),

$$
\text{RR}(A \to B) = \frac{N_{m,A \to B} * \frac{(N_{m,B} + N_{m,0})}{N_m} + N_{f,A \to B} * \frac{(N_{f,B} + N_{f,0})}{N_f}}{N_{m,B} * \frac{N_{m,A \to B} + N_{m,A}}{N_m} \sum_i + N_{f,B} * \frac{N_{f,A \to B} + N_{f,A}}{N_f} N_{i,B}}
$$
(35)

and the male sex-specific relative risk is as specified in Eq. (36),

$$
\text{RR}(A \to B)_m = \frac{\frac{N_{m,A \to B}}{N_{m,A \to B} + N_{m,A}}}{\frac{N_{m,B}}{N_{m,B} + N_{m,0}}}
$$
(36)

and likewise for women.

In this study, we are not interested in identifying inverse comorbidities, i.e. cases where an exposure reduces the risk of a later event. Hence, we define a ROPE that the lower bound of the RR should exclude 1.1. This corresponds to at least a 10% increase in risk. If a combination of diagnoses has a preferred direction and the RR lower bound of that direction excludes 1.1, we say that the co-occurrence was a directional pair.

To compare the relative risk between men and women we subtract the posterior distribution of $\text{RR}_{men}$ and $\text{RR}_{women}$ from each other. If the resulting BCI excludes $(-0.1, 0.1)$, i.e. there should be a difference in risk of at least 10% or more, we note that there is a significant difference in relative risk between men and women.

To compare the difference in directionality strength, we inspected the median of the posterior distribution for the joint direction, and subtracted the posterior distribution of direction men and women, respectively, as shown in Eqs. (37) and (38).

$$\omega_{\text{Men}} = \Pr(A \to B)_{\text{joint}} - \Pr(A \to B)_{\text{Men}},$$
(37)

$$\omega_{\text{Women}} = \Pr(A \to B)_{\text{joint}} - \Pr(A \to B)_{\text{Women}}.$$
(38)

We tested if there was a difference in the variance of the distribution using the F-test. This requires that the distributions are normally distributed. We confirmed this by visual inspection of the density plots (Supplementary Fig. 1). We report the ratio between variances (men compared to women) as the effect size, and the 95% CI.

Literature validation and comparison of co-occurrences with a higher risk in men or women was performed by searching PubMed for articles mentioning either diseases, or, a more specific relevant term. Articles matching were inspected for cohort sizes and to identify any sex-specific estimate of risk or mentions of sex as a risk factor.

**Difference in time between diagnosis**. The time between two diagnoses is computed across all patients that have been diagnosed with both diagnoses. We only look into the directional pairs, defined by an elevated relative risk and pre-ferred direction. We notice that, due to the long follow-up, the distributions have a heavy tail, and are thus not normally distributed. Therefore, we used the two-sided Mann−Whitney $U$ test. Only directional pairs found in both men and women are investigated. Effect sizes are reported as the median difference in time, and the $p$ value is corrected for multiple testing using the BH method. A median difference in time less than zero indicates that the disease transition progress faster in women, and likewise a median difference in time greater than zero indicate that the pro-gression is faster in men.

**Diagnosis trajectories**. We pieced together directional pairs of diagnosis to form multistep trajectories[18,19]. For every pairwise co-occurrence, we iteratively added a diagnosis and counted the number of people following the trajectory in the population. In this particular study, we only investigated trajectories followed by more than 100 people, with a minimum of four diagnoses. Using the disease trajectory framework, we studied two categories of trajectories. First, we investi-gated the directional pairs that had the biggest difference in relative risk between men and women. Second, we selected two diseases, obstructive lung diseases and osteoporosis, which prior studies had found to be underdiagnosed in women and men, respectively. The trajectories were visualized as networks, in which each node represents a diagnosis and the connection between two nodes, the edge, represents a directional link between two diagnoses.

**Reporting summary**. Further information on experimental design is available in the Nature Research Reporting Summary linked to this article.

## Data availability

The study was approved by the Danish Data Protection Agency (ref: 2015-54-0939 and SUND-2017-57) and Danish Health Authority (ref: FSEID-00001627 and FSEID-00003092). Permission to access and analyze data can be obtained following approval from Danish Data Protection Agency and the Danish Health Authority. A reporting summary for this article is available as a Supplementary Information file. Stan (v 2.17)[64], Python (v2.7), and R (v.3.1.3) was used for statistical analysis. Due

to privacy concerns, the provided Supplementary Data only contain estimates for diagnosis and co-occurrences when it has been assigned to at least five men and women.

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

## Acknowledgements
We would like to acknowledge funding from the Novo Nordisk Foundation (grant agreements NNF14CC0001 and NNF17OC0027594).

## Author contributions
D.W. and S.B. conceived the study. S.B. obtained the funding. D.W. and S.B. performed the literature search, figures, study design, and data analysis. D.W., F.K.H.S., P.M., P.B., and S.B. contributed to data interpretation. D.W. and S.B. wrote the initial draft, and D.W., F.K.H.S., P.M., P.B. and S.B. contributed to the final article.

## Additional information

**Competing interests:** The authors declare no competing interests.

