## [Peer Review File · Nature Communications]

Reviewer #1 - Sex and gender-specific medicine (Remarks to the Author):

This is an impressive systematic study of all diseases and disease co-occurrences in a complete European population. The authors claim to analyse sex and gender related differences and partially fulfill this claim. (see below).

The aspects are novel and the study could open novel views on the discussion of sex and gender related diseases. In this way, the findings are of interest to the community. The database is strong and statistical analysis is presented in a convincing manner (more expertise may be needed here for details), figures are well designed.

However, the paper is very descriptive and therefore does not allow for all the relevant conclusions it could. If the authors could at least make some conclusions on whether they are dealing with sex or gender related mechanisms this could sign. improve the paper. However, this important issue is completely lacking.

In the paper, starting in the introduction, sex and gender is not defined and not used in the proper manner, rather in an interchangeable way, what is not correct. It is clear that the distinction between the two mechanisms is not easy but the authors should at least clearly state or discuss whether they believe to detect sex- or gender-related phenomena. In some of their examples, e.g. the inverse relationship between IHD and SV tachycardia in women and men, it may be hypothesized that the difference is due to different awareness of patients and doctors for IHD in women and men. This should at least be discussed. Severity of disease manifestations at hospital presentation, delays between first symptoms and hospital admission could give clues.

Different life span between women and men is not included in the discussion. Fig 2 looks as if a normalization has been undertaken, but it is not clear to me, how.

Reviewer #2 - Population-based analysis and sex-specific traits (Remarks to the Author):

The paper by Westergaard and colleagues examines differences in disease progression among Danish men and women. In terms of good quality data and size, this is as good as it gets. The paper is fairly descriptive and offers little insight into the mechanisms that explain those differences, this is likely my main criticism.

On the technical side, the authors make a number of decisions which I believe could affect the analyses and conclusions and it would be good that they showed that the results are not affected by these.

For instance, excluding data (not clear to me how much, line 407) where the two diagnoses were done the same day would affect the results. Are men and women equally likely to be diagnosed with two conditions the same day? Could the authors repeat the analyses and show that this has no effect on the overall conclusions?

Also, the authors excluded dagger-asterisk pairs, presumably to avoid bias from the clinicians reporting? A sensitivity analyses similar to the one suggested above would be an interesting addition to the paper. Also, how well do the authors' results agree with the dagger-asterisk system? That is, does the dagger->symptom order agree with their results?

Finally, another aspect that worries me is that of power. The paragraph starting in line 96 states a number of directional pairs that are sex-specific, but to what degree it is more likely to find that directionality effect in only one gender because the study has more power to detect that effect in that gender? That is, does the incidence difference in the two sexes for A explain the possibility of finding those directional effects (A->B). My guess is that it does, but it would be good to show that this is not the case.

Reviewer #3 - Population-based health analysis (Bayesian model)(Remarks to the Author):

General comments:

1. The description of the Danish National Patient Registry needs to be expanded to be informative to readers who are not familiar with the Danish health system and what health encounters the registry does and does not include.
2. The authors have made a rather unfortunate choice to present analyses at ICD chapter aggregates which are rather meaningless collections of very different clinical entities or by ICD-10 three digit codes which may either be too detailed (and better aggregated to a clinical entity), reflective of a clinical entity, or too coarse to detect an important clinical entity that can only be determined at the 4 digit level. The three digit ICD-10 codes also are a mix of clinical entities (a disease or injury) and less precise symptoms (e.g. retention of urine) or unspecified residual codes (e.g. other disorders of urinary system). Linked to this comment is a concern that summary stats on 'average' male-female differences by ICD chapter such as in figure 1 are influenced by the level of detail at three digit level ICD happens to have codes rather than the number of entities that have clinical relevance. In the Global Burden of Disease study which the authors refer to in a couple of instances, may provide an alternative classification based on ICD codes that picks up more relevant clinical entities. The GBD list can be expanded upon as it is not exhaustive in listing all specific diseases of clinical relevance but has residual groupings under each disease grouping (e.g.

cardiovascular diseases, chronic respiratory diseases) that mostly follow the ICD chapters but at time split ICD chapters (e.g. separating out acute and chronic respiratory diseases)

3. The choice to limit 'diagnoses' to first occurrence seems reasonable for chronic diseases but seems inappropriate for short duration diseases such as an episode of pneumonia or injuries which can have multiple relevant occurrences and be excluded from the diagnosis trajectories if occurring in a more distant past at a first instance (but still relevant at a second ...or third etc occurrence linked to an incident chronic disease of interest)

Specific comments:

Line 21 Abstract alludes to an analysis of hospital admissions. The (too) brief description of the "registry" also mentions that admissions included outpatient and emergency contacts (line 338). It is unusual to include outpatient contacts as admissions. The description does not clarify what these outpatient contacts include. I'm guessing visits to a medical specialist in a hospital but not GP visits. What about visits to medical specialists outside the hospital? Also, there is no definition of 'emergency contacts'; would that be equivalent to emergency department visits? Based on lines 337-338, it seems that only full-day inpatient and 'half-day' inpatient episodes were included in the analysis. Can authors confirm that? Also, are 'half-day inpatient episodes what elsewhere would be called 'day admissions'?

Line 24 Throughout paper, I find the terminology "diagnosed earlier" a little imprecise. What is measured is a difference in age at first 'diagnosis' (by proxy of age at a first occurrence of a three digit ICD-10 code in inpatient registry). Diagnosed earlier implies a time period between first occurrence of disease and first diagnosis.

Lines 27-29 The abstract is not very informative. For instance, multi-step diagnosis trajectories is undefined jargon. Also, a statement '...uncovered differences in extended longitudinal patterns, for example concerning' does not invite a reader of the abstract to want to read the whole paper. If this is an important finding of the study, why not mention what the differences were and by mentioning 'for example' it is not clear what reader can expect in rest of paper.

Line 72: you define BCI here but are not consistent in its use. For instance, in lines 433 and 437 you mention confidence intervals.

Line 76: "...on average." From looking at fig 1A, I suspect you are making this statement about the average based on the values for each individual ICD-10 3-digit code. As the 'density' of ICD-10 3-digit level codes varies considerably between the ICD chapters relative to the number of clinically coherent categories (some of which have lots of ICD codes at this level; others with just a single code), this may skew some of your comparisons

Line 87: "...confounding factors" For what and why would these factors be confounders?

Line 97: "...and preferred statistical direction". I'm not a great fan of Nature journals' habit of placing the methods section last. It is more problematic in a paper like this in which you introduce many uncommon concepts and analyses. However, in the case of this term, even looking through the

methods section, I see no clear explanation of the meaning of 'preferred'. By whom and how determined?

Line 106: explain what you mean with 'postive deviations'

Line 111: I find fig 3 very hard to interpret. Also, I find ICD-chapter level comparisons not very informative as these contain a heterogeneous set of clinical entities

Line 125: "Risk factors..." Undefined term and a little unusual here, as I would think of smoking, obesity, high blood pressure, air pollution etc as risk factors but that is not what you are alluding to here.

Line 126: "...elevated risk" ...of what?

Line 128: "...in the statistically determined order." Not explained and I have no intuition of what you mean

Line 150: "...diagnosed later..." You mean "diagnosed at an older age"

Line 157: another use of the term 'risk factor'. To me that implies a statement about causality. I think "precedes" is a better alternative

Line 159: "...men mediate..." "male sex mediates", perhaps?

Line 162/3: "trends/trend". Please define what you mean here

Line 170: investigating

Line 172: "relative risk" of what?

Line 180/1: "This analysis..." Referring to what part of your analyses? ...or to all of your analyses presented in this paper?

Line 197: ref 34 seems to have no bearing on this sentence

Line 208/9: you are overstating the claim of being "most comprehensive analysis ...presented so far..."

Line 216: I have not found a clear explanation of how you have dealt with changing diagnostic criteria

Line 220: Many countries see changes in coding practices of hospital diagnoses based on financial reimbursements/incentives. How is that in Denmark?

Line 234: "...women are more likely to abuse drugs..." That seems an odd statement as (illicit) drug abuse is always higher in men than in women. I think you are referring to women more frequently getting an ICD code for drug toxicity, correct?

Line 244: why would small numbers lead to inflated incidence rates or relative risks? There are ways of deriving standard errors for small count rates. While a confidence interval around a small rate

based on a low count of cases may be asymmetrical, it should not bias the mean. Linked to this, I would also query your statement of line 443.

Line 247: Citation of ref 44 is inappropriate. You are probably referring to the use of a geographical hierarchy in the Bayesian meta-regression tool used in GBD but that does not make it a valid comparison with your method.

Line 252: "...an improper prior". You mean an improper prior of a parametric distribution?

Line 302: "...indicators of domestic- and drug abuse". You are not convincing me. What results presented in this paper support that?

Line 312: What is the mechanism through which COPD would lead to osteoporosis? Corticosteroid use? If so, why in men but not in women?

Line 318: "...strongly suggests..." Seems rather a leap to make such an emphatic statement

Line 326/7: This single statement suggesting your results align with GBD is a very loose assertion with nothing to back it up. Either you expand on this and why you think that is the case or you should remove it.

Line 338: "...from before 2002..." Why?

Line 338/9: what is reason for removal of referral diagnoses? I presume it has something to do with avoiding double counting but you are not making that clear.

Lines 379-382: What do you mean with population at risk and why would you add a term with the log of your denominator count?

Line 387: A non-sequitur statement. In a large and small population you can have a low or high incidence rate.

Line 403: "...the weighted mean" Weighted for what? The number of cases per ICD code?

Line 417: why did you decide to sample 5 non-exposed cases? How did you sample?

Line 418/9: "...from the same type of encounter" What do you mean. You have not defined encounter types

Line 436: what is N1? Type: Log RR

Line 450: $\mu_{i,j}$ or did you mean $\eta_{i,j}$ like in your formula?

Line 507: "BH method", an unexplained acronym

Line 508: progresses

Line 698/9: "Of these, 372 pairs2,331 a female-specific diagnosis" This is not reflected in your flow chart

Fig 3: Very difficult to interpret. What is the meaning of the directionality scale? How can you have a proportion greater than 100% (ch 19 and 5 in M). How to interpret the numbers in the boxes and the

colors from the directionality scale? What does it mean when you say “..the boxes indicate the breakdown of the overall co-occurrence figures”?

Line 719: “edges” unexplained jargon

Table 2: are you sure you have almost 100 times greater number of men compared to women for unspecified lump in breast?

Figure 1 A-C: x-axis shows difference between males and females and looks to be expressed as male incidence minus female incidence in graph A. That makes the addition of the male and female signs to the right and left at bottom of graph a little misleading

Fig 2: “pre-screening’ needs to be explained in a footnote to figure

Fig 4: I don’t see θ defined anywhere

Dear Editor,

Thank you for the mail and the three very useful reviews. We would like to thank the reviewers for taking the time to provide such substantial and constructive input. We feel that the manuscript has improved considerably, in particular due to the entire repetition using the Global Burden of Disease. This additional analysis has confirmed the robustness of the previous findings. In a few cases we disagree with the reviewers, but we have tried to clarify the reasoning behind our statements and improved the wording.

We have uploaded the revised manuscript in two versions, one where the changed text is colored green, and one without coloring. All details are described in the point-to-point response below. As you can see we indeed found many of the suggestions relevant, and feel that we have addressed all aspects in the revision and return a substantially improved manuscript.

Søren Brunak

Responses to Reviewers' comments:

Reviewer #1 - Sex and gender-specific medicine (Remarks to the Author):

This is an impressive systematic study of all diseases and disease co-occurrences in a complete European population. The authors claim to analyse sex and gender related differences and partially fulfill this claim. (see below). The aspects are novel and the study could open novel views on the discussion of sex and gender related diseases. In this way, the findings are of interest to the community. The database is strong and statistical analysis is presented in a convincing manner (more expertise may be needed here for details), figures are well designed. However, the paper is very descriptive and therefore does not allow for all the relevant conclusions it could. If the authors could at least make some conclusions on whether they are dealing with sex or gender related mechanisms this could sign. improve the paper. However, this important issue is completely lacking.

Answer: We thank the reviewer for acknowledging the legitimacy of the database and study. We agree with the reviewer that the paper was largely descriptive, with focus on specific gender and sex differences. We have now further amended this by including three specific case stories, and we now also discuss whether a given mechanism could be related to sex and gender in the Discussion. We also note that, by suggestion of reviewer #3, we have included a separate analysis using the Global Burden of Disease fatal and non-fatal categories. This analysis is largely complementary, and we now give results for two different disease classifications and have been amended to the supplementary material. This comparison adds further in terms of specific examples.

In the paper, starting in the introduction, sex and gender is not defined and not used in the proper manner, rather in an interchangeable way, what is not correct. It is clear that the

distinction between the two mechanisms is not easy but the authors should at least clearly state or discuss whether they believe to detect sex- or gender-related phenomena. In some of their examples, e.g. the inverse relationship between IHD and SV tachycardia in women and men, it may be hypothesized that the difference is due to different awareness of patients and doctors for IHD in women and men. This should at least be discussed. Severity of disease manifestations at hospital presentation, delays between first symptoms and hospital admission could give clues. Different life span between women and men is not included in the discussion. Fig 2 looks as if a normalization has been undertaken, but it is not clear to me, how.

Answer: We agree with the reviewer and had not included definitions due to space constraints. We have added to the Introduction a definition of sex and gender in concordance with the WHO definitions. Even if we did not include the definition, we had in fact followed this definition in the work already. We agree that it is extremely important to use the correct definitions of sex and gender, and we have read the text critically for consistency throughout. In the expansion on the Global Burden of Disease, we now comment on the “resolution” of sex and gender related issues in the two different classifications, GBD and ICD. We have also added to the Discussion on the inverse relationships, and we thank the reviewer for providing one hypothesis. Figure 2 presents the workflow showing how we start from 951,509 initial diagnosis co-occurrences and after filtering end with the final numbers reported in the article. There is no normalization applied in Figure 2.

Reviewer #2 - Population-based analysis and sex-specific traits (Remarks to the Author):

The paper by Westergaard and colleagues examines differences in disease progression among Danish men and women. In terms of good quality data and size, this is as good as it gets. The paper is fairly descriptive and offers little insight into the mechanisms that explain those differences, this is likely my main criticism. On the technical side, the authors make a number of decisions which I believe could affect the analyses and conclusions and it would be good that they showed that the results are not affected by these. For instance, excluding data (not clear to me how much, line 407) where the two diagnoses were done the same day would affect the results. Are men and women equally likely to be diagnosed with two conditions the same day? Could the authors repeat the analyses and show that this has no effect on the overall conclusions?

Answer: We thank the reviewer for acknowledging that the data underlying this analysis is of high quality. We agree that the exclusion of events happening on the same day in principle could occur in a sex or gender biased manner. However, the resolution of the Danish National Patient Registry is one day without hour registration, and we therefore cannot resolve the order in which two diagnoses were assigned on the same day. Thus, it is not possible to quantify the temporal relationship further. This has been clarified in the Methods section. One should also factor in that many examinations on the same day often are driven by logistics and waiting lists that may fluctuate from day to day. Therefore, resolving temporal order with hourly precision might not always be relevant or meaningful.

Also, the authors excluded dagger-asterisk pairs, presumably to avoid bias from the clinicians reporting? A sensitivity analyses similar to the one suggested above would be an interesting addition to the paper. Also, how well do the authors' results agree with the

dagger-asterisk system? That is, does the dagger->symptom order agree with their results?

Answer: We have now included the dagger-asterisk pairs in a sensitivity analysis, and found that the results do not change significantly. The analysis has been included in the Supplementary material. We found that our framework generally agrees with the dagger-asterisk system, in which the etiology is diagnosed prior to the manifestation.

Finally, another aspect that worries me is that of power. The paragraph starting in line 96 states a number of directional pairs that are sex-specific, but to what degree it is more likely to find that directionality effect in only one gender because the study has more power to detect that effect in that gender? That is, does the incidence difference in the two sexes for A explain the possibility of finding those directional effects (A->B). My guess is that it does, but it would be good to show that this is not the case.

Answer: As the reviewer mentioned in the starting paragraph with regards to data quality and size, this is as good as it gets. In the full population, there were 48.2% women. The power to detect a direction, A->B, is determined by the total number of patients that has either A->B or B->A. To answer the question from the reviewer, we have plotted the number of men with A->B and B->A and the number of women with A->B and B->A. We found that there is a correlation coefficient of 0.86, and we have added the plot to the Supplementary Materials. Furthermore, we marked the 4,155 directional pairs that the reviewer specifically mentions. We acknowledge that some of them may be due to a lack of power, such as co-occurrences with breast cancer, which has an extremely low incidence in men. However, the results show that this is not the case for the majority of the pairs investigated. Looking only at the 4,155 pairs, the correlation decreases slightly, to 0.80. Reviewer #1 speculates that our observation in the relationship between IHD and SV tachycardia could be due to differences in awareness of doctors and patients – this is exactly the kind of relationship that we *also* wish to identify and point out.

Reviewer #3 - Population-based health analysis (Bayesian model)(Remarks to the Author):

1. The description of the Danish National Patient Registry needs to be expanded to be informative to readers who are not familiar with the Danish health system and what health encounters the registry does and does not include.

Answer: We have expanded the description of the Danish National Patient Registry in the Methods section. We hope that this gives a better insight into its strengths and limitations.

2. The authors have made a rather unfortunate choice to present analyses at ICD chapter aggregates which are rather meaningless collections of very different clinical entities or by ICD-10 three digit codes which may either be too detailed (and better aggregated to a clinical entity), reflective of a clinical entity, or too coarse to detect an important clinical entity that can only be determined at the 4 digit level. The three digit ICD-10 codes also are a mix of clinical entities (a disease or injury) and less precise symptoms (e.g. retention of urine) or unspecified residual codes (e.g. other disorders of urinary system). Linked to this comment is a concern that summary stats on 'average' male-female differences by ICD chapter such as in figure 1 are influenced by the level of detail at three digit level ICD happens to have codes rather than the number of entities that have clinical relevance. In the Global Burden of Disease study which the authors refer to in a couple of instances,

may provide an alternative classification based on ICD codes that picks up more relevant clinical entities. The GBD list can be expanded upon as it is not exhaustive in listing all specific diseases of clinical relevance but has residual groupings under each disease grouping (e.g. cardiovascular diseases, chronic respiratory diseases) that mostly follow the ICD chapters but at time split ICD chapters (e.g. separating out acute and chronic respiratory diseases)

Answer: We agree with the reviewer that the ICD-10 level 3 has some limitations. This is however how the diagnoses have been coded since 1994 – a level 4 analysis would require an extremely large population, even in large countries it may be had to find geographic regions where an unbiased analysis as the one we present would be feasible. However, we have now repeated the analysis using the nonfatal and fatal categories from the Global Burden of Disease study as well. Here, we have used primarily the most specific level, level 4, unless a level 3 code existed only. The results are presented as a supplement to the original findings. In some cases, such as alcoholic cardiomyopathy, the GBD terminology allowed us to pinpoint the exact cause of a difference in cardiomyopathy. In other cases, using only the GBD terminology, we would have missed key findings related to COPD. We have included a discussion of the limitations of both terminologies. We have furthermore added a set of supplementary files that describe whether a GBD category is sex-specific, and which functional system it belongs to. We have limited the functional categories to be the same as those in the ICD-10 terminology. This also makes it possible to highlight the similarities and differences when using a consistent coloring scheme for easy comparison of the corresponding figures for ICD-10 and GBD.

3. The choice to limit ‘diagnoses’ to first occurrence seems reasonable for chronic diseases but seems inappropriate for short duration diseases such as an episode of pneumonia or injuries which can have multiple relevant occurrences and be excluded from the diagnosis trajectories if occurring in a more distant past at a first instance (but still relevant at a second ...or third etc occurrence linked to an incident chronic disease of interest)

Answer: The choice to include the first occurrence of a diagnosis only was made due to technical reasons. It is extremely difficult to determine when a short duration disease is a recurrence, or just repeated due to the patient changing wards (or similar). Often, for non-acute conditions, there are waiting lists at the hospitals. Waiting times fluctuate over the 20 year period, due to political decisions on budgets, prioritization of disease areas like cancer etc. Hence, we did not include recurrences because we felt it could potentially introduce bias and spurious findings. We do concur that recurrence of diagnosis is something that could be very interesting to explore in a sex and gender related context, but it is outside the scope of the current manuscript. Thus, we in principle agree with the reviewer and have added this limitation to the Discussion.

Specific comments:

Line 21 Abstract alludes to an analysis of hospital admissions. The (too) brief description of the “registry” also mentions that admissions included outpatient and emergency contacts (line 338). It is unusual to include outpatient contacts as admissions. The description does not clarify what these outpatient contacts include. I’m guessing visits to a medical specialist in a hospital but not GP visits. What about visits to medical specialists outside the hospital? Also, there is no definition of ‘emergency contacts’; would that be

equivalent to emergency department visits? Based on lines 337-338, it seems that only full-day inpatient and 'half-day' inpatient episodes were included in the analysis. Can authors confirm that? Also, are 'half-day inpatient episodes what elsewhere would be called 'day admissions'?

Answer: We included all types of hospital admissions that are present in the Danish National Patient Registry. We have clarified the patient type and added a more detailed description of the Danish National Patient Registry as already mentioned above. We thank the reviewer for pointing out these inconsistencies and clarifying which parts were unclear.

Line 24 Throughout paper, I find the terminology "diagnosed earlier" a little imprecise. What is measured is a difference in age at first 'diagnosis' (by proxy of age at a first occurrence of a three digit ICD-10 code in inpatient registry). Diagnosed earlier implies a time period between first occurrence of disease and first diagnosis.

Answer: We fully agree with the reviewer, and have now clarified this throughout the manuscript.

Lines 27-29 The abstract is not very informative. For instance, multi-step diagnosis trajectories is undefined jargon. Also, a statement '...uncovered differences in extended longitudinal patterns, for example concerning' does not invite a reader of the abstract to want to read the whole paper. If this is an important finding of the study, why not mention what the differences were and by mentioning 'for example' it is not clear what reader can expect in rest of paper.

Answer: We have attempted to make the abstract more appealing to the general community within the limit of 150 words, as per the *Nature Communications* submission guidelines. The space constraints will not allow for elaborated mention of the case stories and we feel that this is reasonable per the comprehensive cataloging that the paper presents.

Line 72: you define BCI here but are not consistent in its use. For instance, in lines 433 and 437 you mention confidence intervals.

Answer: The Bayesian Credible Interval (BCI) and Confidence Interval (CI) are not the same. The BCI is determined using a fully specified Bayesian model with priors, and is the summation of the posterior distribution that reflects the uncertainty. The confidence interval is the frequentist summation of a model, assuming fixed parameters. Puga et al. 2015 (<https://www.nature.com/articles/nmeth.3368>) provide an excellent introduction to Bayesian statistics, and other comments of relevance for this question can be found at, for instance <https://stats.stackexchange.com/questions/2272/whats-the-difference-between-a-confidence-interval-and-a-credible-interval>.

Line 76: "...on average." From looking at fig 1A, I suspect you are making this statement about the average based on the value for each individual ICD-10 3-digit code. As the 'density' of ICD-10 3-digit level codes varies considerably between the ICD chapters relative to the number of clinically coherent categories (some of which have lots of ICD codes at this level; others with just a single code), this may skew some of your comparisons

Answer: We have repeated the experiment using the GBD categories and found largely a very consistent pattern between GBD and ICD. We have added the GBD incidence

comparison to the Supplementary Materials, and have now included it in the Abstract, Introduction, Results and Discussion.

Line 87: "...confounding factors" For what and why would these factors be confounders?

Answer: We have clarified that the confounding factors mentioned were related to age, admission type (i.e. inpatient versus outpatient), seasonal variance, and changing diagnostic criteria. The latter is a factor that is specific to this data set, due to its 20 year scope.

Line 97: "...and preferred statistical direction". I'm not a great fan of Nature journals' habit of placing the methods section last. It is more problematic in a paper like this in which you introduce many uncommon concepts and analyses. However, in the case of this term, even looking through the methods section, I see no clear explanation of the meaning of 'preferred'. By whom and how determined?

Answer: We agree with the reviewer that the Methods section would be best placed before the Result section. It is, however, an editorial decision. We have revised the Methods section to now include a clear definition of preferred statistical direction.

Line 106: explain what you mean with 'postive deviations'

Answer: We have reworded the sentence. It should now be clear that the intention was to comment on the skewness of the distribution.

Line 111: I find fig 3 very hard to interpret. Also, I find ICD-chapter level comparisons not very informative as these contain a heterogeneous set of clinical entities

Answer: As the reviewer suggested, we have performed the same analysis using the GBD categories and the equivalent figure can be found as Supplementary Figure 4. Although the number of co-occurrences is smaller, we have highlighted some aspects that we did not find in the ICD-10 analysis.

Line 125: "Risk factors..." Undefined term and a little unusual here, as I would think of smoking, obesity, high blood pressure, air pollution etc as risk factors but that is not what you are alluding to here.

Answer: We have clarified that in this context we mean that a risk factor is a prior diagnosis or disease.

Line 126: "...elevated risk" ...of what?

Answer: We have clarified that one of the sexes were at a higher risk of a future diagnosis or disease.

Line 128: "...in the statistically determined order." Not explained and I have no intuition of what you mean

Answer: We have revised the text so that we consistently use preferred statistical direction when referring to the statistically determined order of appearance for two conditions.

Line 150: "...diagnosed later..." You mean "diagnosed at an older age"

Answer: Thank you for pointing this out, the reviewer's interpretation is indeed correct.

Line 157: another use of the term 'risk factor'. To me that implies a statement about causality. I think "precedes" is a better alternative

Answer: We have specifically avoided implying causality, but here we have revised the statement to indicate association and not a proven causal factor.

Line 159: "...men mediate..." "male sex mediates", perhaps?

Answer: We agree with the reviewer that this is a much better formulation.

Line 162/3: "trends/trend". Please define what you mean here

Answer: We have clarified that the trend we talk about is the trend observed at the population level, which is dominated by the trend observed in men.

Line 170: investigating

Answer: We thank the reviewer for pointing out this spelling mistake.

Line 172: "relative risk" of what?

Answer: The relative risk refers to the directional pairs. The definition of directional pairs was given earlier in the Results section. We have repeated the definition in the paragraph to provide the context.

Line 180/1: "This analysis..." Referring to what part of your analyses? ...or to all of your analyses presented in this paper?

Answer: We have clarified that this refers to the trajectory analysis only. Thank you for pointing out this ambiguity.

Line 197: ref 34 seems to have no bearing on this sentence

Answer: We kindly disagree with the reviewer. The reference was added to substantiate the statement that Danish healthcare is of high quality relatively speaking. We acknowledge that the reference might be better placed closer to the statement regarding the high quality healthcare, but this is ultimately an editorial decision as it would go against the "Author Guidelines".

Line 208/9: you are overstating the claim of being "most comprehensive analysis ...presented so far..."

Answer: To our knowledge, no one has performed an analysis across all disorders in a single cohort of seven million people focusing on the comparative aspect between men and women. While the Global Burden of Disease has done a tremendous amount of work on summarizing disease frequencies in an even larger cohort, the work does not include the temporal history to quantify co-occurrences of diagnoses, their directionality, and embedding in trajectories. We have reworded to make clear that the GBD analysis is based on a larger data set but also how our work differs from GBD.

Line 216: I have not found a clear explanation of how you have dealt with changing diagnostic criteria

Answer: This was described in the section regarding confounding from age and season. By selecting a patient that has been hospitalized in the same time period, the matched patient has also been at risk for having the exposure given the same diagnostic criteria.

Line 220: Many countries see changes in coding practices of hospital diagnoses based on financial reimbursements/incentives. How is that in Denmark?

Answer: Danish hospitals have been receiving reimbursement dictated by the assigned diagnostic codes since 2000. We have now added this to the expanded description of NPR.

Line 234: "...women are more likely to abuse drugs..." That seems an odd statement as (illicit) drug abuse is always higher in men than in women. I think you are referring to women more frequently getting an ICD code for drug toxicity, correct?

Answer: Thank you, this has been clarified in the text.

Line 244: why would small numbers lead to inflated incidence rates or relative risks? There are ways of deriving standard errors for small count rates. While a confidence interval around a small rate based on a low count of cases may be asymmetrical, it should not bias the mean. Linked to this, I would also query your statement of line 443.

Answer: We agree with the reviewer that a rare event should not bias the mean, and have updated the text to reflect that what we meant was the interval. There are ways of deriving SEs for small count rates, but these have been developed because it is difficult to get a proper estimate. In this article we address this problem by applying a Bayesian model.

Line 247: Citation of ref 44 is inappropriate. You are probably referring to the use of a geographical hierarchy in the Bayesian meta-regression tool used in GBD but that does not make it a valid comparison with your method.

Answer: We kindly disagree. The GBD make extensive use of hierarchical models and pooling of information between groups. In the GBD case, it is geographical information. In our case, we simultaneously look at the co-occurrence of multiple disorders.

Line 252: "...an improper prior". You mean an improper prior of a parametric distribution?

Answer: We apologize for the improper terminology. What was meant was uninformative priors.

Line 302: "...indicators of domestic- and drug abuse". You are not convincing me. What results presented in this paper support that?

Answer: We have clarified that our hypothesis builds on the observations made from the trajectories regarding injuries, substance abuse, and complications of substance abuse. We emphasize that this likely can be a gender-related mechanism associated with the physical abuse.

Line 312: What is the mechanism through which COPD would lead to osteoporosis? Corticosteroid use? If so, why in men but not in women?

Answer: We have added a sentence describing how corticosteroid use could be related to this trajectory. One hypothesis is that the diagnosis of asthma happens, on average, ten years earlier in men compared to women. This in turn causes men to be more exposed to medication, such as corticosteroids.

Line 318: "...strongly suggests..." Seems rather a leap to make such an emphatic statement

Answer: We think reference 58 backs this wording, but have removed the word “strongly”.

Line 326/7: This single statement suggesting your results align with GBD is a very loose assertion with nothing to back it up. Either you expand on this and why you think that is the case or you should remove it.

Answer: We have added an entire complementary analysis using the GBD categories in the Supplementary material. We can now show that the directionalities in high detail align with the GBD categories, chapter by chapter. Our comparative statement was based literature review, but now we have substantiated it. We have clarified that the alignment relates to these directionalities and not that we have made a detailed quantification of disability-adjusted life years, years lived with disability, or years of life lost. These cannot be readily calculated from ICD-10 and it is important that we limit that our consistency statement to what actually can be compared. We thank the reviewer for leading us into clarifying this aspect, which have improved the manuscript considerably.

Line 338: “...from before 2002...” Why?

Answer: After 2002 there were no longer half-day inpatients. Contact were either inpatient or outpatient.

Line 338/9: what is reason for removal of referral diagnoses? I presume it has something to do with avoiding double counting but you are not making that clear.

Answer: Referral diagnoses are assigned when a patient is referred to a specific department for testing. For instance, a woman may be assigned a referral diagnosis for breast cancer. However, this is not indicative that she has breast cancer, but a suspicion that must be examined in the oncology department. This has also been clarified in the main text.

Lines 379-382: What do you mean with population at risk and why would you add a term with the log of your denominator count?

Answer: The goal was to model the incidence rate. Hence, we wanted to model new cases / population at risk. The population at risk is defined as the population that is at risk of developing the diagnosis under investigation (e.g., women cannot be at risk when investigating prostate cancer). This can be modelled using a log-linear model with a Poisson link, i.e. a rate model:

$$\log(\# \text{new cases} / \text{population at risk}) = b_0$$

$$\Rightarrow \log(\# \text{new cases}) = b_0 + \log(\text{population at risk})$$

Therefore, we added a log-offset to the model.

Line 387: A non-sequitur statement. In a large and small population you can have a low or high incidence rate.

Answer: We agree with the sentiment from the reviewer; low and high incidence rates are not quantifiable. The point in the sentence was, however, to justify the use of the normal prior with an uninformative standard deviation of 3 on the scale. The number of new cases, compared to the population at risk, will be very low in a cohort of seven million people. Therefore, for the Poisson model to properly fit the data the intercept has to be a negative number of a certain magnitude.

Line 403: “...the weighted mean” Weighted for what? The number of cases per ICD code?

Answer: The reviewer is correct. This has been clarified in the text.

Line 417: why did you decide to sample 5 non-exposed cases? How did you sample?

Answer: The choice to sample five matched patients was based on consideration between having enough power to detect a difference, but also having enough suitable controls for each stratification. The five non-exposed patients were selected based on age and having had the same type of hospital encounters within 3 months of time.

Line 418/9: "...from the same type of encounter" What do you mean. You have not defined encounter types

Answer: Encounter types were defined earlier in the Danish Patient Registry description, but we have also added it to the specific sentence to ensure clarity.

Line 436: what is N1? Type: Log RR

Answer: The N1 refers to the normal distribution at 1-alpha/2 confidence. This has been clarified using parenthesis.

Line 450: $\mu_{i,j}$ or did you mean $\eta_{i,j}$ like in your formula?

Answer: This was a typo. Thank you for catching this mistake.

Line 507: "BH method", an unexplained acronym

Answer: Thank you for pointing this out, it has now been clarified in the main text.

Line 508: progresses

Answer: Thank you for pointing out this spelling mistake.

Line 698/9: "Of these, 372 pairs2,331 a female-specific diagnosis" This is not reflected in your flow chart

Answer: We have updated our flowchart and text to be easier to follow (Figure 2).

Fig 3: Very difficult to interpret. What is the meaning of the directionality scale? How can you have a proportion greater than 100% (ch 19 and 5 in M). How to interpret the numbers in the boxes and the colors from the directionality scale? What does it mean when you say "...the boxes indicate the breakdown of the overall co-occurrence figures"?

Answer: We believe that the reviewer has misinterpreted the numbers and the scale. The numbers within each square is the number of co-occurrences from that particular combination of chapters. In the case of ch 19 and 5 for men, there would be 112 directional pairs. The coloring indicates the average directionality between two chapters, in this case whether the directional pairs have a diagnosis starting from ch 19 followed by ch 5, or vice versa. Using the same example, the majority of the directional pairs would start with a diagnosis from ch 19, followed by a diagnosis from ch 5. We have clarified this in the main text and legends and thank the reviewer for point out the lack of clarity.

Line 719: "edges" unexplained jargon

Answer: We have added a brief description of networks to the main text and caption.

Table 2: are you sure you have almost 100 times greater number of men compared to women for unspecified lump in breast?

Answer: We had accidentally switched the numbers. We apologize for the mistake, and have made sure that no such mistake is present in any other tables.

Figure 1 A-C: x-axis shows difference between males and females and looks to be expressed as male incidence minus female incidence in graph A. That makes the addition of the male and female signs to the right and left at bottom of graph a little misleading

Answer: We added the symbols for clarity. In the context of Figure 1A, a negative value indicates that the diagnosis has a higher incidence in women. A positive value indicates that the diagnosis has a higher incidence in men. We believe that the symbols add to the interpretability.

Fig 2: “pre-screening’ needs to be explained in a footnote to figure

Answer: We have added a sentence explaining the pre-screening in the figure caption.

Fig 4: I don’t see θ defined anywhere

Answer: The reviewer is correct. In the rest of the article we use the notation $\Pr(A \rightarrow B)$ instead of θ . We have adapted the figure to match this notation.

Reviewer #1 (Remarks to the Author):

thank you. All concerns have been addressed as far as possible.

Reviewer #2 (Remarks to the Author):

I have no further comments.

Reviewer #3 (Remarks to the Author):

Thank you for a thorough response to review.

I have a few remaining comments:

1. lines 72-74: (a) these are methods statements; (b) confusingly first sentence mentions age-standardised incidence rates but in next sentence you mention that uncertainty was derived from a model with sex and age as covariates.

2. 283-285 you don't make clear why you judge the GBD category of COPD as 'unspecific'. Are you referring to fact that in GBD COPD is defined based on spirometry and that we classify a large proportion of those as 'asymptomatic', particularly at younger ages while you are more likely to just pick up on those with symptomatic COPD?

3. lines 383-396: I'm not convinced by your 'post-hoc' explanation of finding a link between osteoporosis and COPD/asthma in males but not females. Long term use of steroids through inhalers is recommended in asthma but not COPD where steroids tend to be used for shorter periods for 'exacerbations'. Asthma is more prevalent in adult women than in adult men (sex reversal after puberty) and thus osteoporosis due to steroid use in people with asthma would be expected to be higher in females.

4. line 501: you are still stating that in a large population, incidence for any given diagnosis will be low. That still does not make sense to me....

Dear Editor,

Thank you for the mail and the reports. We would like to take the opportunity to once again thank the reviewers for their comments. We are aligned with the comments from Reviewer #3, and have added additional text to the Discussion reflecting this.

As requested, we have highlighted all the changes made in the text of the revised manuscript.

Søren Brunak

Responses to Reviewers' comments:

Thank you for a thorough response to review.

I have a few remaining comments:

1. lines 72-74: (a) these are methods statements; (b) confusingly first sentence mentions age-standardised incidence rates but in next sentence you mention that uncertainty was derived from a model with sex and age as covariates.

Answer: We have altered the sentence, and now refer the reader to the Methods section for a detailed account of the statistical model. Regarding the second question, in a Bayesian model the posterior yields a distribution over the parameters of the model. The predictions, using the posterior distribution of the parameters, then yield an uncertainty regarding a specific combination of strata, in this case the age group and sex. This could, in theory, be summarized to provide an adjusted interval. Here, we take it a step further by weighting the adjusted interval using a standard population that reflects the European population structure (Eurostat 2013 standard population). The exact calculations are detailed in the Methods section under "Diagnosis incidence rates". We do, however, agree that the presentation of this is not sufficiently clear. By removing the lines describing the model, and instead referring the reader to the Methods section, we remove this uncertainty.

2. 283-285 you don't make clear why you judge the GBD category of COPD as 'unspecific'. Are you referring to fact that in GBD COPD is defined based on spirometry and that we classify a large proportion of those as 'asymptomatic', particularly at younger ages while you are more likely to just pick up on those with symptomatic COPD?

Answer: In our analysis, we have looked at some of the components that make up COPD individually (chronic bronchitis and emphysema). As detailed in the trajectory analysis relating to Figure 7 and the case story on Respiratory disorders, there were differences between the components that we would otherwise not have been able to identify if we strictly went with the GBD definition of COPD. Hence, we termed it unspecific because using the ICD-10 codes we were able to analyze some of the specific components. We have now added text that make it clearer why we use the word specific here.

3. lines 383-396: I'm not convinced by your 'post-hoc' explanation of finding a link between osteoporosis and COPD/asthma in males but not females. Long term use of steroids through inhalers is recommended in asthma but not COPD where steroids tend to be used for shorter periods for 'exacerbations'. Asthma is more prevalent in adult women than in adult men (sex reversal after puberty) and thus osteoporosis due to steroid use in people with asthma would be expected to be higher in females.

Answer: We thank the reviewer for his comments regarding the link we have described between COPD and osteoporosis. We largely agree with the reviewers' position. We would like to offer several additional points of clarification. First, the analysis is tempered by the fact that COPD is a substantially under-diagnosed disease (estimated at 50-80% underdiagnosed; (1-2). Even in diagnosed COPD, the percent of spirometrically confirmed COPD is low (approx. 50% of diagnosed COPD patients; (3-4). Thus, we suggest, based upon these and the observations from our submitted manuscript, that patients receiving a diagnosis of COPD are generally more symptomatically severe and would be expected to have received more systemic steroid exposure. Second, recent data suggest that radiographic evidence of moderate to severe emphysema is a risk factor for osteoporosis (5)

More importantly, and as suggested by the reviewer, the COPD phenotype itself carries risks of osteoporosis due to COPD associated frailty, smoking effects on bone metabolism, exercise limitation, etc. In addition, there is an interesting and emerging set of studies showing vitamin D receptor polymorphisms in patients with COPD and osteoporosis (6). The relative impact of these factors would be greatest in males, given the baseline higher (>4 times) level of osteoporosis in women compared to men by age 50.

Regarding gender and osteoporosis in asthma, our findings are in line with earlier studies noting an increase in males (7). We have modified our discussion to reflect the multifactorial nature of osteoporosis in COPD.

4. line 501: you are still stating that in a large population, incidence for any given diagnosis will be low. That still does not make sense to me....

Answer: We have clarified in the text that the prior was chosen as the offset in the linear model will almost always be much bigger, compared to the number of new cases.

REFERENCES:

- 1) Martinez CH, Mannino DM, Jaimes FA, Curtis JL, Han MLK, Hansel NN, Diaz AA. Undiagnosed obstructive lung disease in the United States: Associated factors and long-term mortality. *Ann Am Thorac Soc* 2015;12:1788–1795.
- 2) Çolak Y, Afzal S, Nordestgaard BG, Vestbo J, Lange P. Prognosis of asymptomatic and symptomatic , undiagnosed COPD in the general population in Denmark : a prospective cohort study. *Lancet Respir Med* 2017;5:426–434.
- 3) Arne M, Lisspers K, Ställberg B, Boman G, Hedenström H, Janson C, Emtner M. How often is diagnosis of COPD confirmed with spirometry? *Respir Med* 2010;104:550–556.

- 4) Koefoed MM, Christensen RD, Sondergaard J, Jarbol DE. Lack of spirometry use in Danish patients initiating medication targeting obstructive lung disease. *Respir Med* 2012;106:1743–1748.
- 5) Bon J, Zhang Y, Leader JK, Fuhrman C, Perera S, Chandra D, Bertolet M, Diergaarde B, Greenspan SL, Sciurba FC. Radiographic Emphysema, Circulating Bone Biomarkers, and Progressive Bone Mineral Density Loss in Smokers. *Ann Am Thorac Soc*. 2018 May;15(5):615-621
- 6) Kim SW, Lee JM, Ha JH, Kang HH, Rhee CK, Kim JW, Moon HS, Baek KH, Lee SH. Association between vitamin D receptor polymorphisms and osteoporosis in patients with COPD. *Int J Chron Obstruct Pulmon Dis*. 2015;10:1809-17
- 7) Ji J, Hemminki K, Sundquist K, Sundquist J. Seasonal and regional variations of asthma and association with osteoporosis: possible role of vitamin D in asthma. *J Asthma*. 2010 Nov;47(9):1045-8.

Reviewer #3 (Remarks to the Author):

Response to latest round of comments:

1: fine

2: not a satisfactory answer. Codes of chronic bronchitis and emphysema carry very little additional information due to variations in coding practices. The entity of COPD is the construct of interest and both codes belong to that entity. While in GBD we also make use of e.g. medical claims data based on ICD-coding for COPD, we consider it a suboptimal source in comparison to representative population surveys that used spirometry to identify cases of COPD. In fact we make an upward adjustment to the claims data to reflect a systematic bias towards under-identification of cases in claims data. I don't think you have enough argument to use the word 'unspecific' to qualify the category of COPD used in GBD

3: you don't have me convinced that your post-hoc explanation for your finding of a link between COPD and osteoporosis in men but not women is correct.but that is the nature of post-hoc speculation about a possible underlying mechanism.

4: I do not understand the argument in the answer as it is a grammatically incorrect sentence. However, I no longer see the sentence that triggered my comment.

Dear Editor,

Thank you for the mail and the reports. As requested, all changes to the main document have been done with track changes enabled.

Søren Brunak

Responses to Reviewers' comments:

1: fine

Answer: Thank you.

2: not a satisfactory answer. Codes of chronic bronchitis and emphysema carry very little additional information due to variations in coding practices. The entity of COPD is the construct of interest and both codes belong to that entity. While in GBD we also make use of e.g. medical claims data based on ICD-coding for COPD, we consider it a suboptimal source in comparison to representative population surveys that used spirometry to identify cases of COPD. In fact we make an upward adjustment to the claims data to reflect a systematic bias towards under-identification of cases in claims data. I don't think you have enough argument to use the word 'unspecific' to qualify the category of COPD used in GBD

Answer: Based on the reviewers' arguments and in-depth knowledge of chronic respiratory diseases, we have altered our statement to now highlight that the GBD and ICD-10 use different definitions.

3: you don't have me convinced that your post-hoc explanation for your finding of a link between COPD and osteoporosis in men but not women is correct.but that is the nature of post-hoc speculation about a possible underlying mechanism.

Answer: We agree that it is indeed impossible to justify causality from a retrospective study, and that the effect of sex is something that should be examined in a future prospective study.

4: I do not understand the argument in the answer as it is a grammatically incorrect sentence. However, I no longer see the sentence that triggered my comment.

Answer: We would like to apologize for the incorrect grammar and acknowledge that this did indeed make the sentence difficult to understand.